# ONE-STEP IMAGE-FUNCTION GENERATION VIA CONSISTENCY TRAINING

## ABSTRACT

Consistency models aim to deliver a U-Net generator to map noise to images directly and enable swift inference with minimal steps, even trained in isolation with consistency training mode. However, the U-Net generator requires heavy feature extraction layers for multi-level resolutions and learning convolution kernels with specific receptive fields, resulting in the challenge that consistency models suffer from heavy training resources and fail to generate images with any user-specific resolutions. In this paper, we first validate that training the original consistency model with a small batch size via consistency training mode is pretty unstable, which motivates us to investigate efficient and flexible consistency models. To this end, we propose to use a novel Transformer-based generator to generate *continuous image functions*, which can then be differentially rendered as images with arbitrary resolutions. We adopt implicit neural representations (INRs) to form such continuous functions, which help to decouple the resolution of generated images and the total amount of the parameters generated from the neural network. Extensive experiments on one-step image generation demonstrate that our method greatly improves the performance of consistency models with low training resources and also provides an efficient any-resolution image sampling process.

## 1 INTRODUCTION

Diffusion models Sohl-Dickstein et al. (2015); Song & Ermon (2019; 2020) have achieved remarkable efficacy in synthesizing various signals, including audio Kong et al. (2020); Chen et al. (2020), image Dhariwal & Nichol (2021); Ramesh et al. (2022) and video Harvey et al. (2022); Ho et al. (2022). However, diffusion models rely on an iterative sampling process, leading to slow generation. Consistency model Song et al. (2023) is an emerging family of diffusion models Ho et al. (2020); Song et al. (2020a) that directly map noise to data by maintaining point consistency on ODE trajectory. This unique characteristic enables consistency models to support rapid one-step generation for high-quality samples by design. Consistency models Song et al. (2023) can be trained through two modes: consistency distillation from a pre-trained diffusion model or consistency training in isolation. Different from consistency distillation, consistency training does not rely on the pre-trained diffusion model but utilizes an unbiased estimator to approximate the ground-truth score function Song et al. (2023). Consequently, consistency training emerges as a more flexible and convenient method for training consistency models and shows greater potential in generation tasks.

However, consistency models face challenges due to their substantial training resource requirement and inflexible image generation with fixed resolution. Consistency models rely on a U-Net generator Ronneberger et al. (2015) to map noise to images, while the U-Net involves extensive convolution to extract features and is proved to be less scalable than the Transformer-based generator in diffusion models Peebles & Xie (2023). Our investigation reveals that training consistency models based on a U-Net generator under the consistency training model requires a large batch size. Directly reducing the training batch size to accommodate limited training resources significantly diminishes the generation performance, leading to the generation of non-realistic images, as illustrated in Figure 1 (a). Besides, as depicted in Figure 1 (b), to generate a specific-resolution image, the U-Net needs to denoise a noisy image with the same resolution, causing consistency models can only generate images with fixed resolution once trained on a dataset with specific-resolution images. Therefore, a more efficient and flexible generator is desired for consistency models.

Figure 1: (a) When we train the consistency models on the CelebA dataset with consistency training and a batch size of 4, the generation performance gradually diminishes. (b) The U-Net generates images with the same resolution as the noisy images. Therefore we need to train infinite separate generators at all resolutions if we want to sample images with arbitrary resolutions. (c) Our image function generator treats the images as continuous functions parameterized as the MLPs. With a single generator, it generates a fixed amount of MLP parameters as an image function, which can then be rendered as images with arbitrary resolutions.

In this paper, we propose a novel and efficient generator for consistency models that stabilizes the consistency training process under low training resources and can generate images with any user-specific resolution in the inference phase. Specifically, instead of generating discrete grid representations of images, we treat images as continuous functions and introduce a novel Transformer-based generator Dosovitskiy et al. (2020) that predicts the continuous functions of images as intermediates. These intermediates can then be differentially rendered into images with arbitrary resolution, as depicted in Figure 1 (c). To represent the continuous functions of images, we leverage implicit neural representations Xie et al. (2022) (INRs) that employ Multi-layer Perceptrons (MLPs) to map the coordinates $x \in \mathbb{R}^2$ to corresponding RGB values $y \in \mathbb{R}^3$. Compared to the U-Net generator, our Transformer-based generator exhibits a more stable consistency training process of consistency models with limited training resources. Moreover, the use of differential rendering enables the decoupling of the resolution of the generated images and the total amount of parameters generated from the neural network, leading to efficient sampling of images at arbitrary resolutions.

Additionally, we empirically confirm that consistency training from scratch to generate functions of high-resolution images is challenging and converges slowly. To enhance training efficiency, we carefully design a novel architecture for the function generator, which consists of a feature extraction module to denoise the noisy images, a function prediction module that predicts functions for clean images, and a non-learnable render module to obtain clean images. We then introduce an image reconstruction pre-training task to pre-train the function prediction module, which compels the generator to convert a given clean image to its corresponding INR function, mitigating the optimization challenges in consistency training when predicting the function of the clean image based solely on a given noisy image.

We summarize the contributions of this work as follows:

- Toward efficient and flexible one-step generation for any-resolution images with consistency models, we propose a novel Transformer-based function generator, which first generates image functions with a single inference step, and then renders the image functions to produce images with arbitrary resolutions.

- We carefully design a novel end-to-end architecture for the function generator that simplifies the generation of functions for clean images from given noisy images. This architecture involves a feature extraction module, a function prediction module, and a render module.

- To enhance the consistency training efficiency, we introduce an image reconstruction pre-training task to fortify the function prediction module, enabling the function prediction module to predict functions of images giving clean images. This approach allows consistency training process to concentrate on the image-denoising task, resulting in more realistic image generation and faster convergence of the training process.

- Extensive experiments on one-step image generation under low training resources demonstrate that our methods can generate significantly more realistic images with much less training and inference resources than the original consistency models.

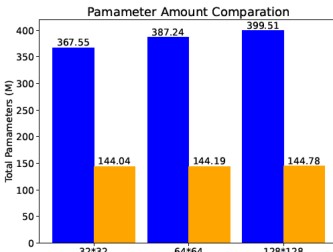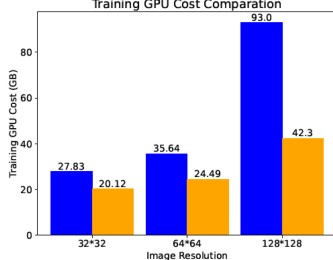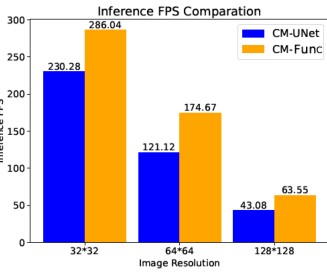

Figure 2: The comparison of the consistency training efficiency with the U-Net generator and our function generator based on image functions, including the parameter amount (left, lower is better), the training GPU costs (middle, lower is better), and inference frame per second (FPS, right, higher is better). We can observe that the increment of the parameter amount of our generator is nearly negligible, while the training GPU cost and inference FPS of our function generator are more efficient and have better scalability than the U-Net generator.

## 2 RELATED WORK

**Diffusion Models.** Diffusion models Sohl-Dickstein et al. (2015); Song et al. (2020a) are emerging topics in the computer vision community. They train a generator to denoise the noise-corrupted data to estimate the score of data distribution and iteratively denoise the data point sampled from the noise distribution to generate new samples. Lots of work are done to accelerate the inference speed of diffusion models, such as faster ODE solver Song et al. (2020a); Lu et al. (2022a;b), predictor-corrector methods Song et al. (2020b) distillation methods Salimans & Ho (2022); Meng et al. (2023) Yin et al. (2024) and rectification Liu et al. (2022; 2023c).

**Consistency Models.** Consistency models Song et al. (2023) is a new type of diffusion model for few-step sampling while maintaining good generation quality. They deliver consistency mapping to map any point in ODE trajectory to its origin, enabling one-step generation. Unlike GAN-based generation models Goodfellow et al. (2014), consistency models do not rely on adversarial optimization and thus avoid the associated training difficulty. Consistency models can be trained either by consistency distillation mode from a pre-trained diffusion model or by consistency training mode with an unbiased estimator to approximate the ground-truth score function. Luo *et al.* Luo et al. (2023) apply consistency distillation to train consistency models in latent space. In this work, we focus on consistency training because it does not rely on an additional pre-trained diffusion model and is more flexible and convenient for training.

**Efficient Diffusion.** Apart from classical U-Net Ronneberger et al. (2015), several works Bao et al. (2023a;b); Peebles & Xie (2023) successfully adopt Vision Transformer Dosovitskiy et al. (2020) as the generator in diffusion models. They Bao et al. (2023a); Peebles & Xie (2023) show that the Transformer generator enjoys better scalability and higher performance in generating high-resolution images in latent diffusion models Rombach et al. (2022) than the U-Net generator. However, these works only explore the simple case of replacing the U-Net generator in the diffusion models with a ViT generator. On the contrary, our work is the first work to deliver a ViT generator to generate INR functions in the novel consistency models that support one-step generation.

**Flexible Image Generation.** Some works have targeted the problem of flexible image generation with any resolution, either based on modifying the diffusion trajectory and model or based on a patch-by-patch assembly manner. Haji-Ali et al. (2023); Zhang et al. (2023) focus on the flexible sampling of U-Net-based diffusion models and enable iteratively generating images with specific resolutions by decoupling the generation trajectory or dynamically adjusting the feature map size. However, these methods rely on operating on the iterative sampling process, which makes them unsuitable for the single-step sampling process with consistency models. The patch-by-patch assembly manner works Chai et al. (2022); Lin et al. (2022) deliver a patch-by-patch strategy to generate patches and assemble the patches into a larger image with particular resolution. However, the output of their generator is still of a fixed resolution, therefore they require multiple inference processes for a larger image and cannot generate images with lower resolution.

**Implicit Neural Representations.** By mapping a coordinate to its corresponding quantity with a neural network (e.g. MLP), INRs have shown great potential in representing complex continuous functions for a lot of natural signals, such as time-serial signals Fons et al. (2022); Szatkowski et al., images Sitzmann et al. (2020b); Skorokhodov et al. (2021); Liu et al. (2023a) and 3D scenes Park et al. (2019); Sitzmann et al. (2020a); Liu et al. (2023b). A lot of works on how to generate INRs fast for unseen signals have been employed, including meta-learning Sitzmann et al. (2020a); Tancik et al. (2021); Liu et al. (2023a); Finn et al. (2017) and hyper-network Chen & Wang (2022); Kim et al. (2023); Zhang et al. (2024).

**Diffusion Models Based on Implicit Neural Representations.** Different from diffusion based on explicit field Zhuang et al. (2023), when adopting diffusion models to high-resolution images or complex 3D signals, existing novel research Dupont et al. (2022); Karnewar et al. (2023); Erkoç et al. (2023); Chen et al. (2024) firstly convert the signals to their corresponding INR functions and then train a diffusion model on these INR functions, enabling generating complex signals efficiently. However, the two-stage training process is inflexible and the error in the first representation stage would greatly affect the performance of the second diffusion stage. On the contrary, our work proposes an end-to-end training framework to generate INRs of signals and requires rendering for only one time when generating one-step inference results.

## 3 INR-BASED CONSISTENCY TRAINING

### 3.1 PRELIMINARIES

Consistency models Song et al. (2023) is a new family of generative models that enables a few-step generation. The core idea of CM is the PF-ODE Song et al. (2020b). Denote the data distribution by $p_{\text{data}}(\mathbf{x})$, and the perturbed distribution is presented as $p_\sigma(\mathbf{x}) = \int p_{\text{data}}(\mathbf{y}) \mathcal{N}(\mathbf{x} \mid \mathbf{y}, \sigma^2 \mathbf{I}) \, d\mathbf{y}$ if we add Gaussian noise $\mathcal{N}(0, \sigma^2)$ with noise level $\sigma$ to the data. Then, the PF-ODE presented in Karras *et al.* Karras et al. (2022) is formulated as:

$$\frac{d\mathbf{x}}{d\sigma} = -\sigma \nabla \log p_\sigma(\mathbf{x}) \approx -\sigma \boldsymbol{s}_\phi(\mathbf{x}, \sigma), \tag{1}$$

where $\boldsymbol{s}_\phi(\mathbf{x}, \sigma) \approx \nabla \log p_\sigma(\mathbf{x})$ is the *score function* Song et al. (2020b) of $p_\sigma(\mathbf{x})$. Here, as in Song et al. (2023); Karras et al. (2022), $\sigma$ is defined as $\sigma \in [\sigma_{\min}, \sigma_{\max}]$, where $\sigma_{\min}$ is a small positive number to ensure $p_{\sigma_{\min}}(\mathbf{x}) \approx p_{\text{data}}(\mathbf{x})$ and $\sigma_{\max}$ is a large positive number such that $p_{\sigma_{\max}}(\mathbf{x}) \approx \mathcal{N}(\mathbf{0}, \sigma_{\max}^2 \mathbf{I})$.

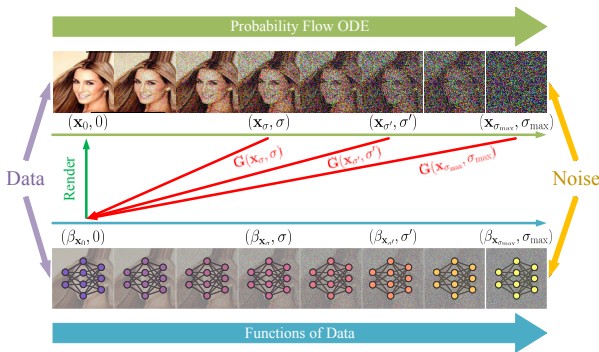

Figure 3: Given a probability-flow ODE that smoothly converts data to noise, we learn a denoising neural network $G$ to map any point (e.g., $\mathbf{x}_\sigma$, $\mathbf{x}_{\sigma'}$, and $\mathbf{x}_{\sigma_{\max}}$) on the ODE trajectory to the continuous function of the origin (e.g., $\beta_{\mathbf{x}_0}$) for generative modeling, which can then be differentially rendered as the original data with arbitrary resolutions.

Solving the PF-ODE from noise level $\sigma$ to $\sigma_{min}$ in Eq. 1 indeed establishes a bijective mapping from a noisy data sample $\mathbf{x}_\sigma \sim p_\sigma(\mathbf{x})$ to the real data sample $\mathbf{x}_{\sigma_{\min}} \sim p_{\sigma_{\min}}(\mathbf{x}) \approx p_{\text{data}}(\mathbf{x})$. This mapping $\boldsymbol{f}^* : (\mathbf{x}_\sigma, \sigma) \mapsto \mathbf{x}_{\sigma_{\min}}$ is defined as a *consistency function* in Song *et al.* Song et al. (2023). By the definition, the consistency function satisfies the *boundary condition* $\boldsymbol{f}^*(\mathbf{x}, \sigma_{\min}) = \mathbf{x}$. To approximate the consistency function with boundary condition, a consistency model $\boldsymbol{f}_\theta(\mathbf{x}, \sigma)$ is parameterized as:

$$\boldsymbol{f}_\theta(\mathbf{x}, \sigma) = c_{\text{skip}}(\sigma)\mathbf{x} + c_{\text{out}}(\sigma)\boldsymbol{F}_\theta(\mathbf{x}, \sigma), \tag{2}$$

where $\boldsymbol{F}_\theta(\mathbf{x}, \sigma)$ is a free-form denoising neural network with parameter $\boldsymbol{\theta}$, while $c_{\text{skip}}(\sigma)$ and $c_{\text{out}}(\sigma)$ are differential functions so that $c_{\text{skip}}(\sigma_{\min}) = 1$ and $c_{\text{out}}(\sigma_{\min}) = 0$.

The consistency function has the property of *self-consistency*: the outputs are consistent for arbitrary pairs of $(\mathbf{x}_\sigma, \sigma)$ that belong to the same PF-ODE trajectory, which can be formulated as:

$$\boldsymbol{f}(\mathbf{x}_\sigma, \sigma) = \boldsymbol{f}(\mathbf{x}_{\sigma'}, \sigma'), \forall \sigma, \sigma' \in [\sigma_{\min}, \sigma_{\max}]. \tag{3}$$

Figure 4: The architecture for our image-function generator. It contains three modules: the feature extraction module that contains several encoder blocks to extract features on the input noisy images and gives data tokens as a condition to guide the generation of the image function; the function prediction module that contains several decoder blocks to predict the INR parameters for image functions with given data token from the previous module; the non-learnable render module that multiples the INR parameters with coordinates and finally render as images.

Therefore, consistency models can be trained by enforcing the self-consistency property with a consistency loss between the results denoised from the $i^{th}$ noise level and the $(i+1)^{th}$ noise level:

$$\mathcal{L} = \mathbb{E}\left[d\left(\boldsymbol{f_\theta}\left(\mathbf{x}_{\sigma_{i+1}}, \sigma_{i+1}\right), \boldsymbol{f_{\theta^-}}\left(\hat{\mathbf{x}}^\phi_{\sigma_i}, \sigma_i\right)\right)\right],\tag{4}$$

where $d(\cdot,\cdot)$ is a metric function such as $\ell_2$ metric or learned perceptual image patch similarity (LPIPS) metric Zhang et al. (2018) while $\boldsymbol{\theta^-} \leftarrow \mu\boldsymbol{\theta^-} + (1-\mu)\boldsymbol{\theta}$ is the target model parameter updated with the exponential moving average (EMA) of the parameter $\boldsymbol{\theta}$ and EMA decay rate $\mu$. $\hat{\mathbf{x}}^\phi_{\sigma_i}$ is derived from $\mathbf{x}_{\sigma_{i+1}}$ by solving the PF-ODE in the reverse direction for a single step:

$$\hat{\mathbf{x}}_{\sigma_i} = \mathbf{x}_{\sigma_{i+1}} - (\sigma_i - \sigma_{i+1})\sigma_{i+1}\nabla_\mathbf{x}\log p_{\sigma_{i+1}}(\mathbf{x})\big|_{\mathbf{x}=\mathbf{x}_{\sigma_{i+1}}}.$$

To estimate the unknown score function $\nabla_\mathbf{x}\log p_{\sigma_{i+1}}(\mathbf{x})$, Song *et al.* Song et al. (2023) propose *consistency training* that employs an approximation $\hat{\mathbf{x}}_{\sigma_i} = \mathbf{x} + \sigma_i\mathbf{z}$ with the same $\mathbf{x} \sim p_\text{data}(\mathbf{x})$ and $\mathbf{z} \sim \mathcal{N}(\mathbf{0}, \boldsymbol{I})$ to calculate $\mathbf{x}_{\sigma_{i+1}} = \mathbf{x} + \sigma_{i+1}\mathbf{z}$. Therefore, the consistency training objective $\mathcal{L}_\text{CT}$ can be defined as:

$$\mathcal{L}_\text{CT} = \mathbb{E}\left[d\left(\boldsymbol{f_\theta}\left(\mathbf{x} + \sigma_{i+1}\mathbf{z}, \sigma_{i+1}\right), \boldsymbol{f_{\theta^-}}\left(\mathbf{x} + \sigma_i\mathbf{z}, \sigma_i\right)\right)\right].\tag{5}$$

## 3.2 GENERATE IMAGE FUNCTIONS AND SAMPLE ANY-RESOLUTION IMAGES

Our modification focuses on the free-form denoising neural network $\boldsymbol{F_\theta} : (\mathbf{x}, \sigma) \rightarrow \mathbf{x}_d$, which is typically a U-Net model Song & Ermon (2019); Ronneberger et al. (2015) that gets noisy image $\mathbf{x} \in \mathcal{R}^{C \times H \times W}$ and noise level $\sigma \in \mathcal{R}$ as input. As shown in Figure 1 (b), the U-Net model applies heavy feature extraction at the different resolution of the image and finally outputs a denoised image $\mathbf{x}_d \in \mathcal{R}^{C \times H \times W}$. We find that the U-Net can only generate the output image with exactly the same resolution as the input image, which is not flexible enough to scale to high-resolution images with limited training resources.

Therefore, we seek a more efficient and more noise-robust method to generate high-resolution images Dupont et al. (2022); Rahaman et al. (2019); Skorokhodov et al. (2021). We show our pipeline in Figure 3. As in INR methods Sitzmann et al. (2020b); Liu et al. (2023a); Chen & Wang (2022), rather than discrete grid representation, we consider images as continuous functions, which can be parameterized as neural networks, e.g. MLPs Dupont et al. (2022). Specifically, we consider an image as a collection of paired coordinates and RGB values $\{(\mathbf{c}, \mathbf{y})\}_{H \times W}$ and fit an MLP $\boldsymbol{M_\beta}$ with learnable parameter $\beta$ to map the coordinates $\mathbf{c} \in \mathcal{R}^2$ to its corresponding RGB values $\mathbf{y} \in \mathcal{R}^3$:

$$\boldsymbol{M_\beta}(\mathbf{c}) = \mathbf{y}.\tag{6}$$

To obtain smooth super-resolution performance and eliminate artifacts, we follow Zhang et al. (2024) to apply variational coordinates to sample the coordinates $\mathbf{c}$.

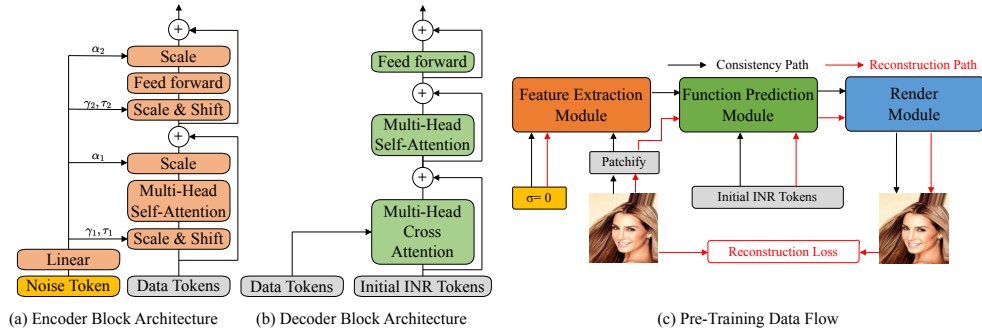

(a) Encoder Block Architecture    (b) Decoder Block Architecture      (c) Pre-Training Data Flow

Figure 5: (a) & (b) The detailed implementation of encoder and decoder block (Norm is not shown). (c) Pre-training data flow. We set the noise level to 0, skip the feature extraction module, and optimize the Function Prediction Module to predict the image function for clean images with reconstruction loss. The black path is the data flow when consistency training the whole model while the red path is the pre-training of the function prediction module with the image reconstruction task.

Note that each image can be considered as an image function $M_\beta$ and holds its own parameter $\beta = \{\mathbf{W}^t, \mathbf{B}^t\}_{t=0}^{t=T-1}$ for an MLP with totally $T$ layers. Generating a denoised image is equivalent to generating the INR parameter $\beta$ of the image function corresponding to the denoised image:

$$\boldsymbol{G}_\theta(\mathbf{x}, \sigma) = \beta, \tag{7}$$

where $\boldsymbol{G}_\theta$ is our proposed function generator and its architecture is introduced in Section 3.3.

Once the parameter $\beta$ is obtained, the denoised image $\mathbf{x}_d$ can be reconstructed at arbitrary resolution by querying the RGB values with any specific coordinates $\{\mathbf{c}\}_{H \times W}$. As a result, we parameterize the free-form neural network $\boldsymbol{F}_\theta$ with a generator that generates image function as intermediate:

$$\boldsymbol{F}_\theta(\mathbf{x}, \sigma, \{\mathbf{c}\}) = \boldsymbol{M}_{\boldsymbol{G}_\theta(\mathbf{x}, \sigma)}(\{\mathbf{c}\}) = \mathbf{x}_d. \tag{8}$$

Note that each layer of MLP can be formulated as:

$$\mathbf{c}^{t+1} = \mathtt{act}(\mathbf{W}^t \mathbf{c}^t + \mathbf{B}^t), \tag{9}$$

where $\mathtt{act}$ is the activation function. The whole forward process of $M_\beta(\mathbf{c})$ is entirely differential if the activation function $\mathtt{act}$ is differential, which enables the end-to-end backward process after calculating consistency training loss, as shown in Eq. 5.

Compared with the U-Net generator that generates images with the same resolution as the input images, our function generator only generates the INR parameters $\beta$ of the image function that has a fixed size and does not scale with the input image resolution. When scaling to the larger resolution image, our pipeline only needs to query the image function with finer-grained coordinates $\{\mathbf{c}\}$, which greatly decreases the parameters amount, training time, and memory cost of the whole pipeline and provides a flexible any-resolution image sampling process.

### 3.3 INR GENERATOR DESIGN

In this part, we introduce the detailed architecture of our function generator. As presented in Figure 4, it contains three major modules: a feature extraction module, a function prediction module, and a render module.

**Feature Extraction Module.** The feature extraction module is utilized to extract features from noisy images and output the data tokens to guide the generation of the image function for the denoised images. We mainly follow DiT Peebles & Xie (2023) and deliver adaLN-Zero Transformer blocks to form a feature extraction encoder. As presented in Figure 5 (a), we adopt a linear layer to regress the dimension-wise scale and shift parameters $\alpha$, $\gamma$, and $\tau$ from noise level embedding.

**Function Prediction Module.** The function prediction module is designed for generating the INR parameters $\beta$ for the image function based on the image feature extracted from the feature extraction

Table 1: Table for the one-step image generation performance. The number in the name means the training resolution.

| Dataset | Models | FID (↓) | SFID (↓) | IS (↑) | P (↑) | R (↑) |
|---------|--------|---------|----------|--------|-------|-------|
| Cifar10-32 | CM-UNet | 32.87 | 20.94 | 6.16 | **0.595** | 0.23 |
| | **CM-Func** | **28.87** | **19.81** | **6.92** | 0.52 | **0.30** |
| CelebA-64 | CM-UNet | 54.41 | 72.86 | 1.77 | 0.43 | 0.021 |
| | **CM-Func** | **29.49** | **45.10** | **2.08** | **0.78** | **0.094** |
| CelebA-128 | CM-UNet | 89.46 | 158.64 | 1.40 | 0.46 | 0 |
| | **CM-Func** | **69.3** | **124.22** | **1.66** | **0.46** | **0.002** |
| LSUN Church-128 | CM-UNet | 58.33 | 88.33 | 1.82 | 0.31 | 0.007 |
| | **CM-Func** | **34.94** | **86.28** | **2.42** | **0.33** | **0.053** |
| LSUN Classroom-128 | CM-UNet | 65.18 | 84.69 | 2.54 | 0.41 | 0.026 |
| | **CM-Func** | **57.96** | **76.31** | **2.83** | **0.50** | **0.033** |

module. Before training, we randomly initialize the learnable INR tokens according to the shape of the parameters $\beta$. As shown in Figure 5 (b), we design a decoder block that adopts multi-head cross attention to fuse the data feature with INR tokens and predict the INR parameters for the image function of specific denoised images. Note that we follow Chen *et al.* Chen & Wang (2022) to use a grouping strategy to improve the efficiency and scalability of our function prediction module.

**Render Module.** After obtaining INR parameters $\beta$ for the image functions, we need to differentially render images to ensure the whole pipeline is differential. As discussed before, we form a continuous image function as an MLP with parameter $\beta$ corresponding to a specific image. With given resolution $H \times W$, we sample a coordinate list $\{(\frac{i}{H}, \frac{j}{W})\}_{i,j}$ where $i \in [0, H)$ and $j \in [0, W)$ and query the RGB value at each coordinate, which finally can be reshaped to form a complete image. There is no learnable parameter in the render module since the INR parameter $\beta$ is generated by the function prediction module.

### 3.4 Benefit from Image Reconstruction Pre-training

We notice that predicting the image function for a clean image from a noisy image is pretty difficult if not impossible, due to the large search space of the INR parameter $\beta$. Therefore, we propose an image reconstruction task to pre-train the function prediction module. We hope the feature extraction module focuses on transforming the noise image feature into the clean image feature while the function prediction module focuses on transforming the clean image feature into its image function.

Therefore, we design an image reconstruction pre-training task for the function prediction module. as shown in Figure 5 (c). Specifically, during the image reconstruction pre-training task, we skip the feature extraction module and set the noise level $\sigma$ as 0 so that the input image are neither perturbed with noise nor denoised by the feature extraction module. The pipeline is downgraded to a model (only the function prediction module contains learnable parameters) that transforms an input clean image to its corresponding image function with specific INR parameters $\beta$. We then optimize such a model with image reconstruction loss:

$$\mathcal{L}_{\text{rec}} = \text{MSE}(M_{\boldsymbol{G}_\theta(\mathbf{x}, \sigma=0)}(\{c_i\}), \mathbf{x}),$$  (10)

where $\text{MSE}$ denotes the mean square error loss.

After pre-training the function prediction module, we train the whole model (including the feature extraction module and function prediction module) with the consistency training objective shown in Eq. 5, which enables our model to generate new images with given random noise. We will show that the image reconstruction pre-training task helps to make the consistency training process converge faster.

## 4 Experiments

In the experiment section, we first provide a detailed comparison between the original U-Net generator and our function generator in the one-step image generation task based on consistency models under the low-training resource, denoted as *CM-UNet* and *CM-Func* respectively. Then we provide

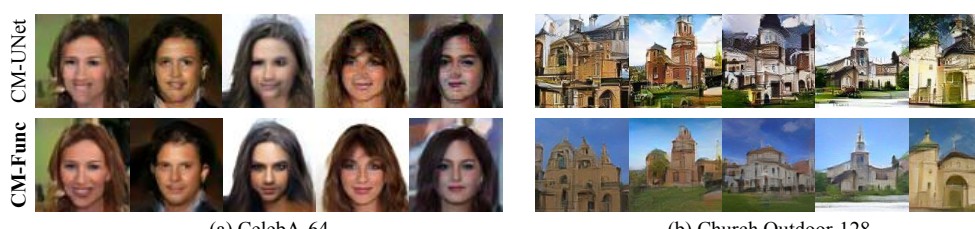

<p style="text-align:center">(a) CelebA-64        (b) Church Outdoor-128</p>

Figure 6: Results from (a) the CelebA dataset with resolution 64 (a) and from (b) the LSUN Church Outdoor with resolution 128. All corresponding images are generated from the same initial noise.

more discussion and ablation studies to illustrate the advantages of function generator when sampling images with different resolutions compared to other popular generators, such as UViT Bao et al. (2023a) and DiT Peebles & Xie (2023).

### 4.1 SETTING

**Datasets.** We show the performance for the unconditional image generation on two popular image datasets: CelebA Liu et al. (2015) with resolution 64 and 128 and LSUN dataset Yu et al. (2015) with resolution 128. We also show that our pipeline can be applied to the class-conditional image generation with the Cifar10 dataset Krizhevsky et al. (2009). We mainly evaluate the models by sampling images with corresponding resolutions.

**Hyper-parameters.** Following the setting of training the consistency models as in Song *et al.* Song et al. (2023), we use the following default hyper-parameters for all datasets unless otherwise stated: $\sigma_{\min} = 0.002$, $\sigma_{\max} = 80$. We use a low batch-size training strategy to evaluate the performance of consistency models under the low-training resource setting. We set the number of encoder blocks N = 8 and the number of decoder blocks M = 6. We adopt MLP with a depth of 5 and width of 256, with ReLU activation, and positional embedding as our image function. For detailed hyper-parameters for consistency models and optimization processes, we present more details in Appendix A.

For the pre-training task, we use the same dataset as the consistency training task and follow Chen *et al.* Chen & Wang (2022) to optimize the models with Adam optimizer Kingma & Ba (2014) and learning rate $1e - 4$ for 30 epochs. All results are reported for models with the pre-training task unless otherwise discussed.

**Metric.** We first compare the efficiency of consistency models with the U-Net generator and function generator. Then we report the quantitative generation results according to Frechet Inception Distance (FID) Heusel et al. (2017), Sliding Fréchet Inception Distance (sFID) Szegedy et al. (2016), Inception Score (IS) Salimans et al. (2016), Precision (P) Kynkäänniemi et al. (2019), and Recall (R) Kynkäänniemi et al. (2019). We follow Song et al. (2023) and Dhariwal & Nichol (2021) to generate 50000 images for credible scores.

We also define a new metric, the total denoising distance, to reflect the denoising quality of our generators during training. The total denoising distance is simply calculated as the L2 difference between the denoised result from the noisy image and the original clean image:

$$d^i_{\text{denoising}} = |\boldsymbol{f}_\theta \left( \mathbf{x}_{\sigma_i}, \sigma_i \right) - \mathbf{x}_0|.$$

Though this metric does not reflect the generation performance in the test phase, it means the denoising ability of the generator during training. More details for this metric are in Appendix B.

### 4.2 RESULTS

**Model efficiency.** We quantitatively compare the efficiency of our function generator and the baseline U-Net generator on the unconditional image generation task to show that our model is more efficient. We report the total parameters, the GPU cost, and the inference FPS in Figure 2. We verify that our model has fewer total parameters, less training GPU cost, and higher inference FPS.

Apart from the absolute quantitative value, we also observe that when increasing the resolution of images (from Cifar10-32 to CelebA-64 and then to CelebA-128), the increment of the total param-

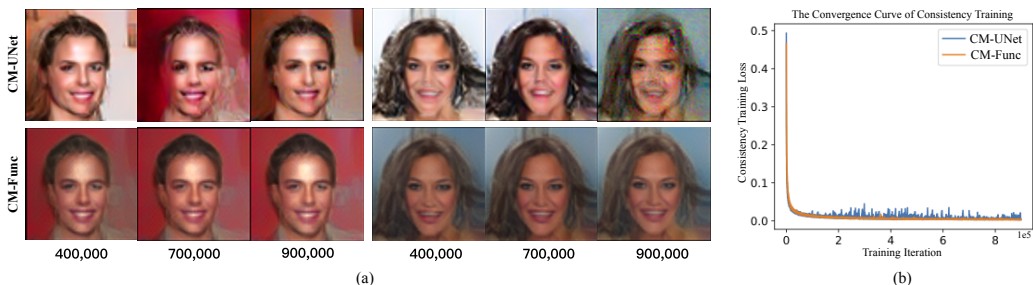

(a)                                                                          (b)

Figure 7: (a) Visual results are generated from the two initial noises by models with different optimization iterations. (b) The convergence curve for the consistency training. The results indicate that the U-Net is unstable while our model is much more stable when keeping training.

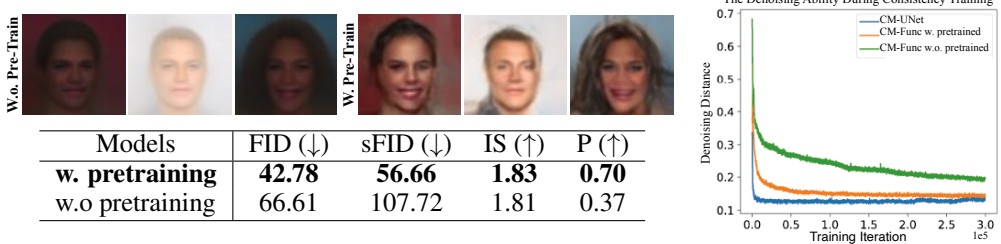

| Models | FID ($\downarrow$) | sFID ($\downarrow$) | IS ($\uparrow$) | P ($\uparrow$) |
|---|---|---|---|---|
| **w. pretraining** | **42.78** | **56.66** | **1.83** | **0.70** |
| w.o pretraining | 66.61 | 107.72 | 1.81 | 0.37 |

Figure 8: Top Left: Visual results of optimizing models with or without pre-training task for 50,000 iterations. Bottle Left: Alabtion of Pre-training task on CelebA-64 with 400,000 training iterations. Right: Denoising ability during training. Pre-training task greatly improves the denoising ability.

eter and GPU cost of our model is much less than the U-Net or even negligible, which verifies that our model has better scalability than U-Net when scaling to image with higher resolution.

**Generation performance.** We demonstrate the quantitative generation performance in Table 1. We can observe that the generation performance on the Cifar10, CelebA, and LSUN datasets beats the baseline U-Net in terms of all evaluation metrics, while our model has much fewer model parameters and enjoys faster training, faster inference, and flexible employment.

We also present some visual generation results based on the CelebA dataset with resolution 64 and the LSUN dataset with resolution 128 in Figure 6. We can clearly observe that given exactly the same initial noise, our model can generate much more realistic than the baseline consistency model with the U-Net generator. More visual and quantitative results are presented in Appendix D.

## 4.3 DISCUSSIONS

**Training oscillates for Consistency models with U-Net generator.** We go deeper into the failing case when training consistency models with U-Net and low batch size 32 and show the results in Figure 7. We find that during the training of the consistency model with U-Net with a low batch size (32 for one single A6000 GPU), the generation results tend to be corrupted and fail to be denoised. On the contrary, the results from the consistency model with our function generator are much more consistent during training and we can find that these results are clean. We also present the consistency training objective convergence curve for the training consistency model in Figure 7 (b). The result shows that the consistency training process with the U-Net generator is pretty unstable and oscillating. In contrast, the process with our function generator is much more stable and consistent.

**Pre-training improves the convergence of the consistency training with the function generator.** We compare the total denoising distance metric of training consistency models with our function generator with or without pre-training task and show the result in Figure 8. We also show the generation performance at 400,000 iterations on the CelebA-64 dataset by ablating the pre-training task in the Bottle Left Table in Figure 8. The results show that our pre-training reconstruction task greatly improves the denoising ability of our function generator during the early stage of consistency training and leads to better generation performance. We also verify that our pre-training task is very efficient and costs much less time compared with the consistency training process. More quantitative results for the efficiency of the pre-training task are shown in Appendix C.

Table 2: Table for efficiency and accuracy of different generators on CelebA-64 dataset.

| Models | Efficience | | Accuracy | | | |
| --- | --- | --- | --- | --- | --- | --- |
| | Any-Resolution Sampling | Multi-Resolution Sampling FPS (↑) | FID (↓) | sFID (↓) | IS (↑) | P (↑) |
| CM-UNet | ✗ | 83.64 | 54.41 | 72.86 | 1.77 | 0.43 |
| CM-UViT | ✗ | 83.58 | 40.14 | 41.47 | 1.90 | 0.68 |
| CM-DiT | ✗ | 96.61 | 25.52 | 37.27 | 2.00 | 0.81 |
| **CM-Func** | ✓ | **447.02** | 29.49 | 45.10 | 2.08 | 0.78 |

**Flexible sampling image resolution leads to a more efficient sampling process.** A comprehensive experiment on the CelebA dataset shown in Table 2 is conducted to evaluate the efficiency and generation quality. We evaluate the efficiency by measuring the FPS to sample 10000 image signals, each of which requires 3 resolutions ($32 \times 32, 64 \times 64, 128 \times 128$, totally 30000 images). Only our function generator supports sampling any-resolution images with one model, while other generators require training multiple separate models at each resolution. Therefore, the multi-resolution sample FPS for those generators that generates fixed-resolution images is calculated as $\frac{30000}{\sum_{i=1}^{10000} T_{32}^i + \sum_{i=1}^{10000} T_{64}^i + \sum_{i=1}^{10000} T_{128}^i}$, where $T_k^i$ is the average time that generates $i^{th}$ image with resolution $k$. Since the function generator only needs to generate one image function for each image, the multi-resolution sample FPS for our function generator is calculated as $\frac{30000}{\sum_{i=1}^{10000} (T^i + T_{R32} + T_{R64} + T_{R128})}$, where $T^i$ is the average time that generates $i^{th}$ image functions, and $T_{R32}, T_{R64}, T_{R128}$ are the time to render the image function to image with 32/64/128 resolutions (which is nearly negligible compared to $T^i$). The results indicate that our function generator achieves a much higher multi-resolution sampling FPS. In contrast, other generator needs to deliver different models to separately denoise input noisy images, which leads to a slower sampling process.

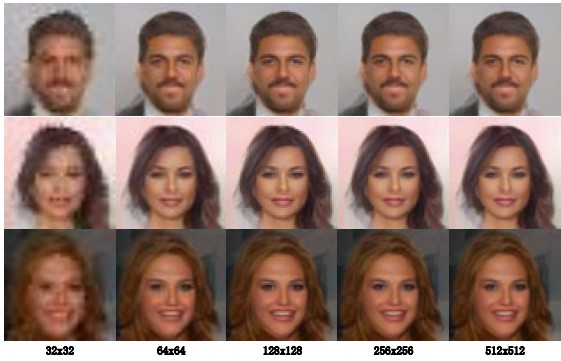

Figure 9: After training on the CelebA-64 dataset, our pipeline supports sampling at any resolution while the results at all resolutions remain clean and realistic.

For generation quality, we find that our function generator greatly beats U-Net and is comparable with Transformer-based generators. We show the generation results with different sampling resolutions in Figure 9. The results indicate that the sampling results at all resolutions remain clean and realistic. We also find that even trained with lower-resolution images, our function generator is still better than the U-Net generator. In addition, we find that our image functions have a better interpolation performance than the linear interpolation. See more quantitative and visualization results in Appendix D.2 and Appendix D.3.

## 5 CONCLUSION

In this paper, we explore efficient consistency training generation for consistency models. We observe that the U-Net generator is resource-intensive and inflexible. Reducing batch size for single GPU use harms performance. To address this, we introduce a Transformer generator that generates image functions parameterized as INR and then renders them into images of any resolution. We design a new end-to-end function predictor that simplifies generating clean image functions from noisy images. Additionally, we enhance training efficiency with an image reconstruction pre-training task. Extensive experiments show our method produces more realistic images with fewer resources than the original consistency models and also provides an efficient any-resolution image sampling process. A limitation of this paper lies in that INRs for image functions are global representations of signals, and have poor representation ability of the local semantic information, which makes our pipeline hard for generation with finer details.

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
