## A    DETAILED EXPERIMENT SETTINGS

In this section, we first present the detailed hyper-parameters that are used to train the consistency models (Section A.1) and the optimization processes (Section A.2). And then we show the detailed architecture of the U-Net that we compare in the main experiments (Section A.3).

### A.1    MORE HYPER-PARAMETERS IN CONSISTENCY MODELS

In this part, we present more details about the hyper-parameters in consistency models Song et al. (2023). We mainly follow the original setting by consistency models Song et al. (2023).

The hyper-parameters in Eq. 2 that parameterized the boundary condition are defined as:

$$c_{\text{skip}}(\sigma) = \frac{\sigma_{\text{data}}^2}{(\sigma - \sigma_{\min})^2 + \sigma_{\text{data}}^2},$$

$$c_{\text{out}}(\sigma) = \frac{\sigma_{\text{data}}(\sigma - \sigma_{\min})}{\sqrt{\sigma_{\text{data}}^2 + \sigma^2}},$$

where $\sigma_{\text{data}} = 0.5$ and $\sigma_{\min} = 0.002$.

In our experiments, we utilize the LPIPS metric Zhang et al. (2018) to compose $d(\cdot, \cdot)$ as in Eq. 4 and Eq. 5 while interpolating the images to resolution $224 \times 224$. The noise level $\sigma_i \in [\sigma_{\min}, \sigma_{\max}]$ are discretized into $N$ sub-intervals with the following equation:

$$N(k) = \left\lceil \sqrt{\frac{k}{K}\left((s_1 + 1)^2 - s_0^2\right) + s_0^2} - 1 \right\rceil + 1,$$

where $s_0, s_1$ are the minimal or maximal number of sub-intervals, while $k$ and $K$ are the current and total training step. For all unconditional image generation tasks, $s_0 = 2$ and $s_1 = 150$. Following Karras *et al.* Karras et al. (2022), the exact value of $\sigma_i$ is defined as:

$$\sigma_i = \left(\sigma_{\min}^{\frac{1}{\rho}} + \frac{i-1}{N(k)-1}\left(\sigma_{\max}^{\frac{1}{\rho}} - \sigma_{\min}^{\frac{1}{\rho}}\right)\right)^{\rho},$$

where $i \in [1, N(k)]$ and $\rho = 7$.

### A.2    MORE TRAINING DETAILS OF OUR EXPERIMENTS

Unless otherwise stated, we follow Peebles *et al.* Peebles & Xie (2023) and Chen *et al.* Chen & Wang (2022) to split the $32 \times 32$ and $64 \times 64$ images with patch size 4, and the $128 \times 128$ images with patch size 8. We set up the number of encoder block $N = 8$ and the number of decoder block $M = 6$. All of the models are optimized with Rectified Adam optimizer Liu et al. (2019) under the learning rate $5e - 4$.

Due to the limitation of training resources, the models for unconditional image generation tasks are optimized with a batch size of 32 on a single 48 GB A6000 GPU. We optimize the models on the Cifar-10 dataset for 400,000 iterations and the models on the CelebA dataset and LSUN datasets for 900,000 iterations to ensure the models converge.

### A.3    DETAILED ARCHITECTURE OF THE U-NET

In this part, we present the detailed architecture of the baseline U-Net. Specifically, we follow the U-Net architecture as in Song *et al.* Song et al. (2023) and Dhariwal *et al.* Dhariwal & Nichol (2021). The detailed architecture hyper-parameters are defined as follows:

- Attention is applied at three different resolutions, including $32 \times 32$, $16 \times 16$, and $8 \times 8$.

- The time embedding is generated with a learnable MLP that is composed of two linear layers whose embedding dimension is 1024. The activation function in the MLP is SiLU. Then the time embedding works as an input for every layer of the U-Net.

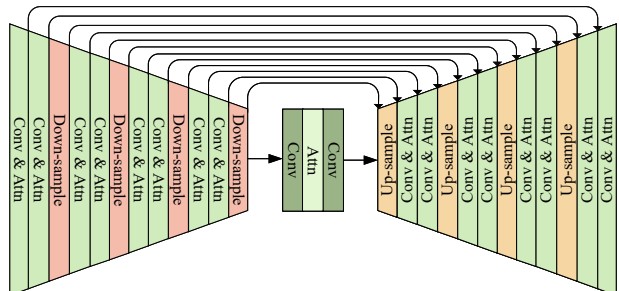

Figure 10: The detailed architecture of baseline U-Net. It contains heavy convolution and attention at different resolutions. The input contains the input noisy images and the time embeddings.

- The channel multiplication has four levels: [1,2,3,4], while in each level, we have 2 resident blocks for the unconditional image generation tasks.

- The dimension of the channel in the resident blocks is set to 256.

- For the unconditional image generation task, we do not use scale and shift norm in the residual blocks.

- The number of the attention head in both down-sample blocks and up-sample blocks is set to 4, while the channel of the head is set to 64.

- Dropout is not applied in our setting.

- The detailed architecture is shown in Figure 10.

## B    THE TOTAL DENOISING DISTANCE METRIC

Note that we follow Eq. 4 to enforce the denoised results of $\mathbf{x}_{\sigma_{i+1}}$ and $\mathbf{x}_{\sigma_i}$ to be close with consistency training objective:

$$d_{\mathrm{CT}}^i = \boldsymbol{f}_\theta \left( \mathbf{x}_{\sigma_{i+1}}, \sigma_{i+1} \right) - \boldsymbol{f}_\theta \left( \mathbf{x}_{\sigma_i}, \sigma_i \right).$$

We can achieve good image denoising only if $d_{\mathrm{CT}}^i$ is small for all $\mathbf{x}_{\sigma_i}$ in the PF-ODE trajectory and we have $\boldsymbol{f}_\theta \left( \mathbf{x}_{\sigma_i}, \sigma_i \right) \approx \mathbf{x}_0$. Therefore, we can define a total denoising distance to reflect the quality along PF-ODE trajectory of our denoising model during training:

$$d_{\mathrm{denoising}}^i = |\sum_{j=0}^i d_{\mathrm{CT}}^j| = |\boldsymbol{f}_\theta \left( \mathbf{x}_{\sigma_i}, \sigma_i \right) - \mathbf{x}_0|,$$

which can be simply calculated as the difference between the denoised image and the original clean image. We can use this metric to monitor the denoising ability of the consistency model during training.

## C    TRAINING EFFICIENCY ANALYSIS

We evaluate the total training time for different models on the celebA-64 dataset to show the efficiency of our model pipeline and training mechanism. As shown in Table 3, the time consumed by the pre-training tasks only accounts for 5% of the total training time. This is because 1) the pre-training task for reconstruction needs much fewer iterations to converge, 2) the pre-training is applied solely to the Function Prediction Module, which accounts for less than half of the total model parameters. Therefore, our pre-training task is very efficient and the extra training cost is almost negligible compared with the heavy cost for consistency training.

Table 3: Table for training efficiency on CelebA-64 dataset.

|  | Total Training Time (GPU * Hours) for CelebA-64 |
| --- | --- |
| CM-UNet | 302.5 |
| CM-DiT | 207.5 |
| CM-UViT | 212.5 |
| CM-Func w.o. pre-training | 223.75 |
| CM-Func w. pre-training | 223.75 + 12.9 |

Table 4: Table for the generation performance on Cifar10-32 and CelebA-64 dataset in terms of image quality assessment metrics.

|  | Models | NIQE ($\downarrow$) | CLIPIQA ($\uparrow$) | MUSIQ ($\uparrow$) | MANIQA ($\uparrow$) |
| --- | --- | --- | --- | --- | --- |
| Cifar10-32 | CM-UNet | 23.003 | 0.516 | 17.043 | 0.105 |
|  | **CM-Func** | **22.264** | **0.520** | **17.143** | **0.107** |
| CelebA-64 | CM-UNet | 6.810 | 0.465 | 21.198 | 0.210 |
|  | **CM-Func** | **6.503** | **0.549** | **22.230** | **0.220** |

## D MORE EXPERIMENT RESULTS

In this part, we present more experiment results to illustrate the performance of our function generator compared to the U-Net generator (Section D.1). We also adapt two more Transformer-based generators that were originally used in diffusion models to the consistency model pipeline and compare the performance between the Transformer-based generators and the U-Net generator (Section D.2). Finally, we show that our function generator enjoys the advantage of generating images with user-specific resolutions compared with the classical Transformer-based generators (Section D.3).

### D.1 MORE RESULTS

We include more quantitative metrics for no-reference image quality assessment to evaluate the performance of our function generator and the baseline U-Net. Specifically, we use the pyiqa package to evaluate these four metrics, i.e., NIQE Mittal et al. (2013), CLIPIQA Wang et al. (2023), MUSIQ Ke et al. (2021), and MANIQA Yang et al. (2022). We follow the default setting to evaluate NIQE, CLIPIQA, and MUSIQ on 50000 generated images and evaluate MANIQA on 1000 generated images for efficiency. The results are presented in Table 4, which demonstrates that the quality of the images generated by our function generator is better than those by the U-Net generator.

We also present the visual results for consistency models trained on Cifar10-32 dataset Krizhevsky et al. (2009) in Figure 11, the visual results for consistency models trained on CelebA-64 dataset Liu et al. (2015) in Figure 12, the visual results for consistency models trained on CelebA-128 dataset Liu et al. (2015) in Figure 13, the visual results for consistency models trained on LSUN-128 dataset Yu et al. (2015) in Figure 14 and Figure 15.

On the Cifar10 dataset, the visual results are not so good. We owe this problem to the limited supervised signals of the Cifar10 dataset. The Cifar10 dataset contains images with low resolution ($32 \times 32$) and can provide very limited supervised signals to predict good image functions. However, even though the visual generation results are not so good, we can observe the difference between the results from the two generators, while the results from the U-Net are more broken and contain more noise. On the contrary, the results from our image generator tend to be cleaner and more vivid. These results show that our image generator performs better than the U-Net generator even on images with pretty low resolution.

On the CelebA-64 and the CelebA-128 datasets, as presented in Figure 12 and Figure 13, we can observe a more obvious broken case of the U-Net generator, which further confirms our conclusion. On the contrary, the consistency model with our function generator tends to generate more clear, realistic, diverse images.

For the LSUN Church-128 datasets, as presented in Figure 14, we find that the U-Net generator tends to generate more artifacts, which prevents the U-Net from generating clearer and more realistic

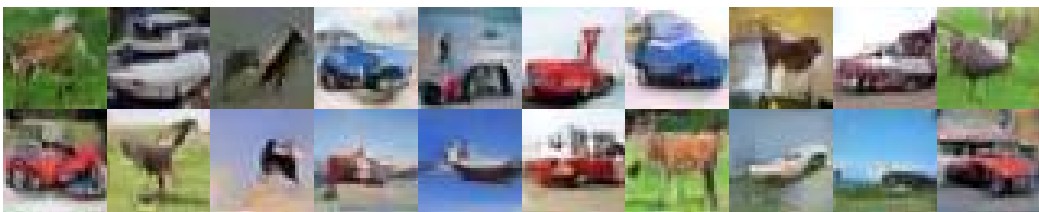

(a) Results from U-Net generator on Cifar10-32 dataset

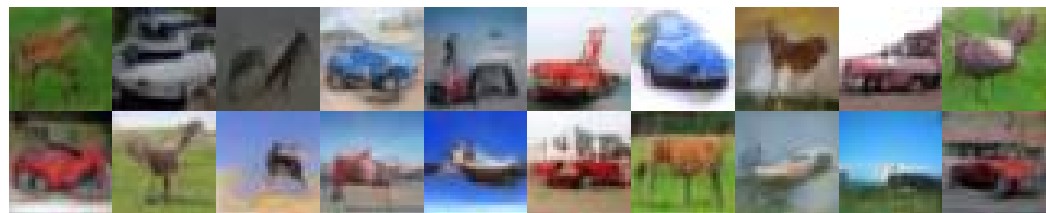

(b) Results from function generator on Cifar10-32 dataset

Figure 11: Visual results from U-Net generator (a) and function generator (b) trained from the Cifar10-32 dataset, while images in the same location are generated from the same initiated noises. Even though these visual generation results are not so good, we can find the difference between the results from the two generators, while the results from the U-Net are more broken and contain more noise. On the contrary, the results from our function generator tend to be cleaner and more vivid.

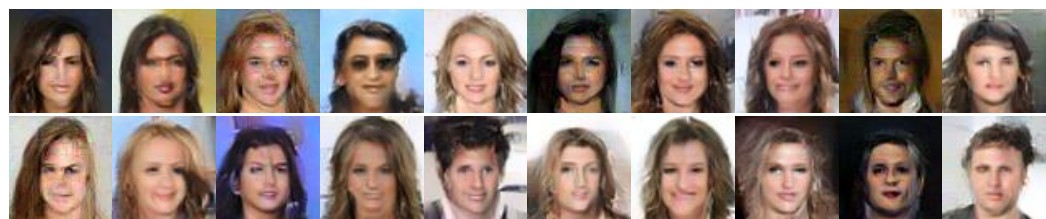

(a) Results from U-Net generator on CelebA-64 dataset

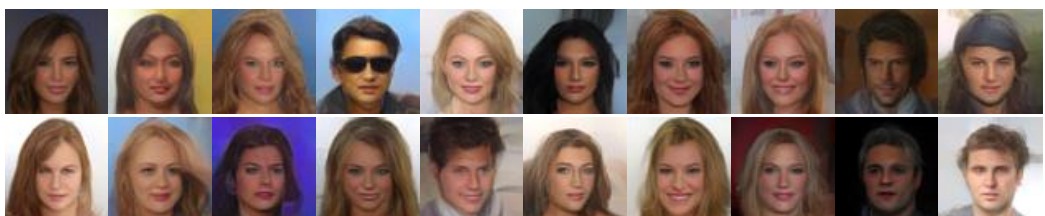

(b) Results from function generator on CelebA-64 dataset

Figure 12: Visual results from U-Net generator (a) and function generator (b) trained from the CelebA-64 dataset, while images in the same location are generated from the same initiated noises. We can observe that the results from the U-Net generator are still noisy. On the contrary, the consistency model with our function generator tends to generate more clear, realistic, diverse images.

images. For the LSUN Classroom-128 datasets, we show the visual results in Figure 15. On such a complex scene dataset, we find that the U-Net also generates more meaningless artifacts. Even though our function generator does not generate very clear details due to the low training batch, it at least generates rough but clear images.

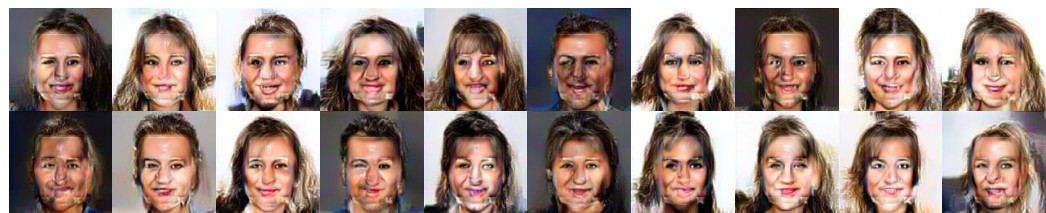

(a) Results from U-Net generator on CelebA-128 dataset

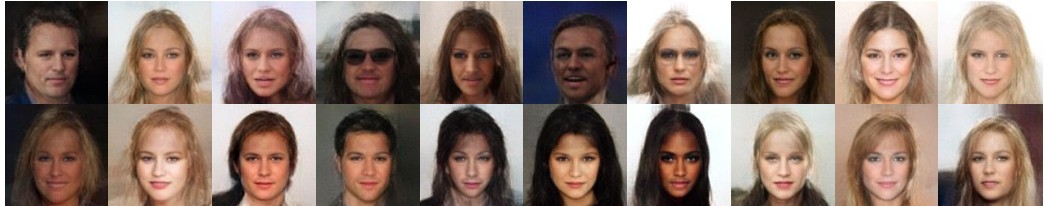

(b) Results from function generator on CelebA-128 dataset

Figure 13: Visual results from U-Net generator (a) and function generator (b) trained from the CelebA-128 dataset, while images in the same location are generated from the same initiated noises. We can observe that the results from the U-Net generator are broken due to the training oscillation. On the contrary, the consistency model with our function generator tends to generate more clear, realistic, diverse images.

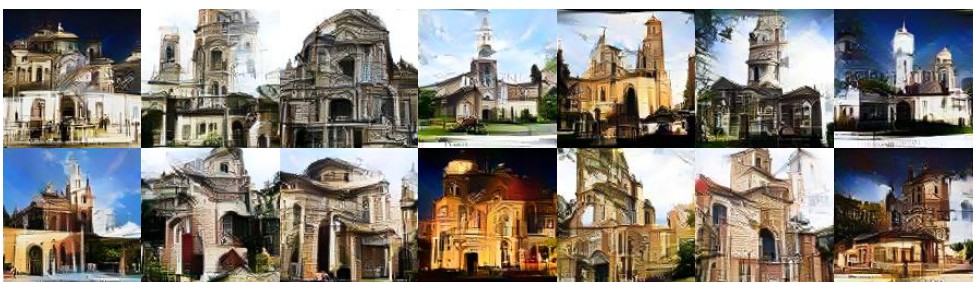

(a) Results from U-Net generator on LSUN Church-128 dataset

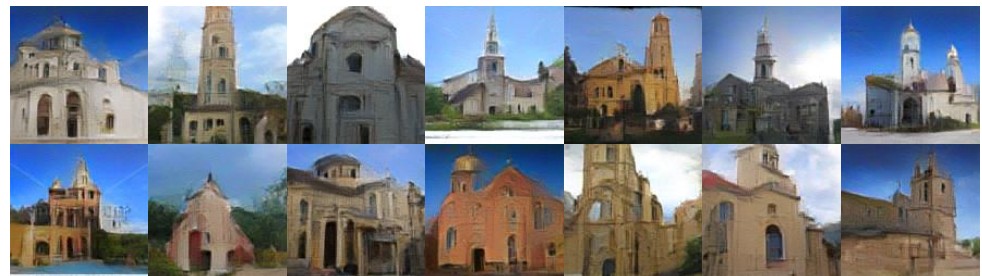

(b) Results from function generator on LSUN Church-128 dataset

Figure 14: The visual results from U-Net generator (a) and function generator (b) trained from the LSUN Church-128 dataset, while images in the same location are generated from the same initiated noises. We can observe that the results from the U-Net generator contain lots of artifacts. On the contrary, the consistency model with our function generator tends to generate more clear, realistic, diverse images.

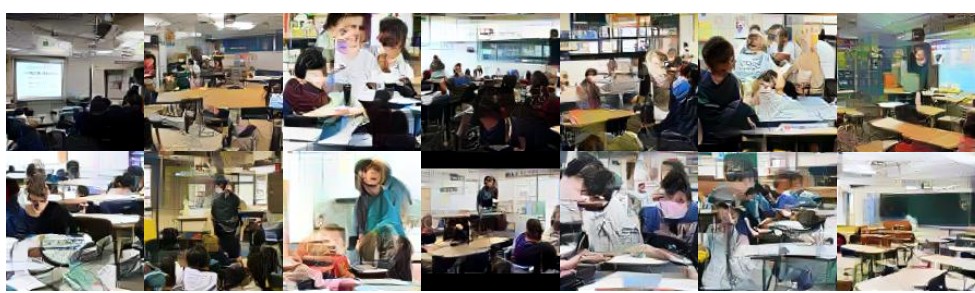

(a) Results from U-Net generator on LSUN Classroom-128 dataset

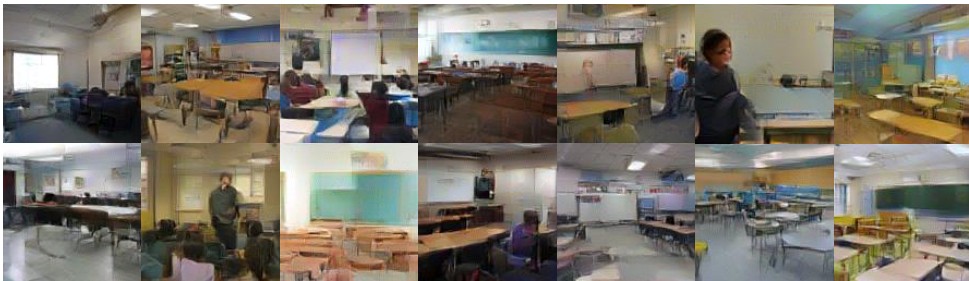

(b) Results from function generator on LSUN Classroom-128 dataset

Figure 15: The visual results from U-Net generator (a) and function generator (b) trained from the LSUN Classroom-128 dataset, while images in the same location are generated from the same initiated noises. We can observe that the results from the U-Net generator contain lots of artifacts. On the contrary, the consistency model with our function generator tends to generate more clear, realistic, diverse images.

## D.2    COMPARISON WITH OTHER GENERATORS

We adapt two more Transformer-based generators that were originally used in diffusion models to replace the U-Net generator in consistency models. One is U-ViT by Bao *et al.* Bao et al. (2023a) and the other one is DiT by Peebles *et al.* Peebles & Xie (2023).

### D.2.1    SETTING

We build these two generators that have a close amount of learnable parameters with our function generator. As a result, the parameter amount and GPU cost of DiT and UViT are very close to our function generators. The detailed architectures are presented as follows.

For U-ViT, the total depth is set to 15 so that the whole model contains 7 layers for the encoder, 1 layer for middle transformation, and 7 layers for the decoder. The skip connection is applied between the corresponding layers of the encoder and decoder. Same as our function generator, we set the patch size as 4 for $64 \times 64$ images and 8 for $128 \times 128$ images, the embedding dimension as 768, number of heads as 12. As for DiT, the same hyper-parameters (depth, patch size, embedding dimension, number of heads) are set to ensure they have a close amount of total parameters.

All models are optimized with the same iterations and the same optimizer as shown in Section 4.1. We evaluate the performance of Transformer-based generators and our function generator on the CelebA dataset with resolution $64 \times 64$.

We use extensive quantitative metrics to evaluate the performance of different generators, including FID Heusel et al. (2017), sFID Szegedy et al. (2016), IS Salimans et al. (2016), Precision Kynkäänniemi et al. (2019), Recall Kynkäänniemi et al. (2019), NIQE Mittal et al. (2013), CLIPIQA Wang et al. (2023), MUSIQ Ke et al. (2021), and MANIQA Yang et al. (2022). We follow the default setting to evaluate all metrics on 50000 generated images, except MANIQA which is evaluated on 1000 generated images for efficiency.

Table 5: Table for the generation performance of consistency models with different generators on CelebA-64 dataset. The best results are marked as bold and the second results are marked with underline.

| Dataset | Models | FID ($\downarrow$) | sFID ($\downarrow$) | IS ($\uparrow$) | precesion ($\uparrow$) | Recall ($\uparrow$) |
|---|---|---|---|---|---|---|
| CelebA-64 | CM-UNet | 54.41 | 72.86 | 1.77 | 0.43 | 0.021 |
| | CM-UViT | 40.14 | 41.47 | 1.90 | 0.68 | 0.020 |
| | CM-DiT | **25.52** | **37.27** | 2.00 | **0.81** | **0.10** |
| | CM-Func | 29.49 | 45.10 | **2.08** | 0.78 | 0.094 |

Table 6: Table for the generation performance for different generators on CelebA-64 dataset in terms of image quality assessment metrics. The best results are marked as bold and the second results are marked with underline.

| | Models | NIQE ($\downarrow$) | CLIPIQA ($\uparrow$) | MUSIQ ($\uparrow$) | MANIQA ($\uparrow$) |
|---|---|---|---|---|---|
| CelebA-64 | CM-UNet | 6.810 | 0.465 | 21.198 | 0.210 |
| | CM-UViT | 6.564 | 0.516 | **22.805** | 0.194 |
| | CM-DiT | 6.666 | **0.561** | 22.139 | **0.234** |
| | CM-Func | **6.503** | 0.549 | 22.230 | 0.220 |

Table 7: Table for the generation performance of consistency models with different training resolutions. The results are evaluated on the CelebA dataset at the resolution of 128.

| Models | Training Resolution | FID ($\downarrow$) | sFID ($\downarrow$) | IS ($\uparrow$) | P ($\uparrow$) | R($\uparrow$) |
|---|---|---|---|---|---|---|
| CM-UNet | 128 | 89.46 | 158.64 | 1.40 | 0.46 | 0 |
| CM-Func | 128 | 69.30 | 124.22 | 1.66 | 0.43 | 0.002 |
| | 64 | 76.93 | 105.90 | 1.98 | 0.18 | 0.0011 |

### D.2.2 RESULTS

**Comparable with other Transformer-based generators.** We present the quantitative results of consistency models with different generators in Table 2, Table 5 and and Table 6. We find that since our function generator is built based on Transformer architecture, it enjoys close generation performance with other Transformer-based generators. In the meanwhile, all generators based on Transformer have a better performance than the U-Net generator, which proves that Transformer as a basic architecture, is more stable and flexible than U-Net in the consistency model pipeline. Furthermore, compared with these two Transformer-based generators, our function generator enjoys the advantage of generating images with arbitrary resolution, which we will discuss in section D.3.

### D.3 SUPPORT GENERATING IMAGES WITH USER-SPECIFIC RESOLUTIONS

### D.3.1 MODIFICATION TO INFERENCE PHASE

With very little modification on the inference process, our pipeline supports one-step image generation with a user-specific resolution during the inference phase,

When we re-examine the inference process of consistency models with our function generator, we find that the main trouble that prevents our models from generating arbitrary-resolution images lies on Eq. 2:

$$f_{\boldsymbol{\theta}}(\mathbf{x}, \sigma) = c_{\text{skip}}(\sigma)\mathbf{x} + c_{\text{out}}(\sigma)\boldsymbol{F}_{\boldsymbol{\theta}}(\mathbf{x}, \sigma),$$

where the input noisy images $\mathbf{x}$ and the denoised images from the neural network generator $\boldsymbol{F}_{\boldsymbol{\theta}}(\mathbf{x}, \sigma)$ must have the same resolution so that they can be added. However, during the inference phase, $c_{\text{skip}}$ is a pretty small number which makes the input noisy images have little effect on the final results. Therefore, we simply modify the Eq. 2 by omitting the $c_{\text{skip}}(\sigma)\mathbf{x}$ term:

$$f_{\boldsymbol{\theta}}(\mathbf{x}, \sigma) = c_{\text{out}}(\sigma)\boldsymbol{F}_{\boldsymbol{\theta}}(\mathbf{x}, \sigma).$$

Note that we completely de-couple the relationship between the resolution of the input noisy images and the output denoised images. The resolution of the images used to train the consistency models

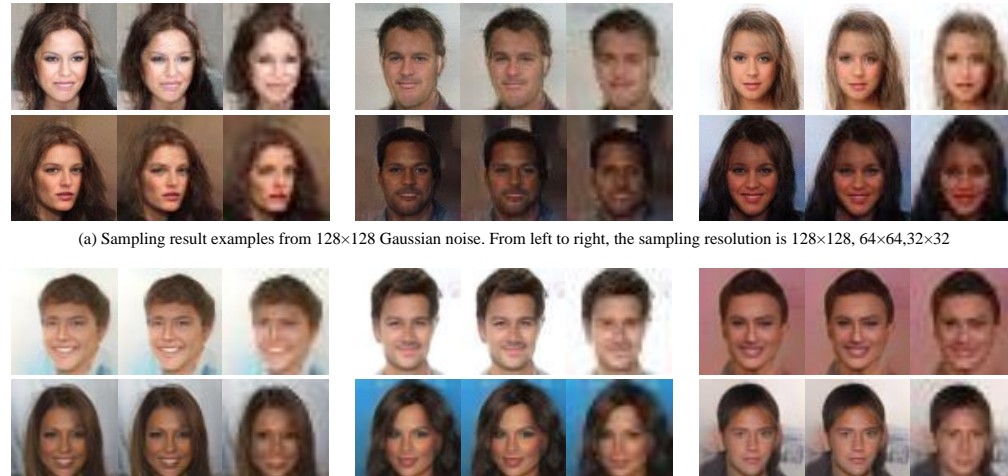

(a) Sampling result examples from 128×128 Gaussian noise. From left to right, the sampling resolution is 128×128, 64×64,32×32

(b) Sampling result examples from 64×64 Gaussian noise. From left to right, the sampling resolution is 128×128, 64×64,32×32

Figure 16: The sampling result examples from $128 \times 128$ Gaussian noise (a) and $64 \times 64$ Gaussian noise (b). From left to right, the three images have resolution $128 \times 128$, $64 \times 64$, and $32 \times 32$. We can observe that the consistency model with our function generator can generate images with arbitrary resolution, while the images at all resolutions remain clean and realistic. We can also find that when sampling images with $128 \times 128$ resolution, the results from $128 \times 128$ Gaussian noise are more realistic than those from $64 \times 64$, which indicates that a larger noise space leads to better generation performance.

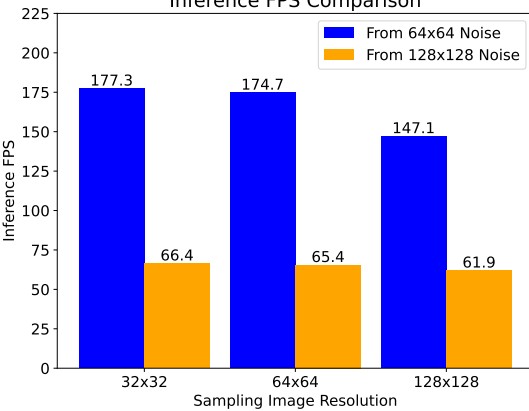

Figure 17: The inference FPS comparison of the consistency models inferring from $64 \times 64$ noise and $128 \times 128$ noise. The result shows that a smaller noise space leads to faster inference.

with our function generator only affects the noise space that we need to denoise. For example, training on $64 \times 64$ images indicates that we need to denoise $64 \times 64$ Gaussian noise vectors in the inference phase while training on $128 \times 128$ images indicates that we need to denoise $128 \times 128$ Gaussian noise vectors in the inference phase. A larger noise space typically implies more diverse image generation because more information can be extracted and stored in the neural network. On the contrary, a faster sampling speed can be obtained with a smaller noise space. We show the performance of our inference phase in Section D.3.2.

### D.3.2 RESULTS

**Flexible sampling image resolution.** We show that the consistency model with our function generator can generate images with arbitrary resolution in Figure 16. We can observe that the results of denoising from the noisy images at all resolutions remain clean and realistic. We can also find that when sampling images with $128 \times 128$ resolution, the results from $128 \times 128$ Gaussian noise are more

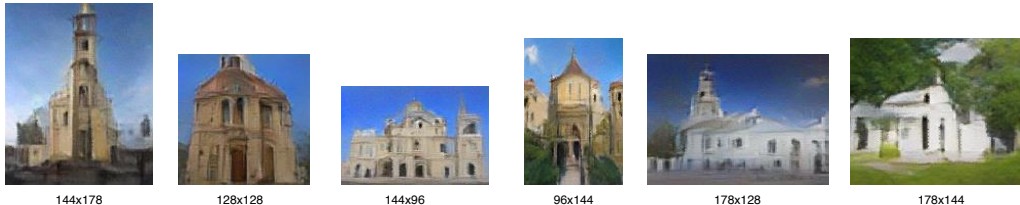

Figure 18: This figure shows some non-squared samples with different height-width ratios from the CM-Func model trained on the LSUN Church-128 dataset. By simply querying at different coordinates, we can achieve image generation with any height-width ratio.

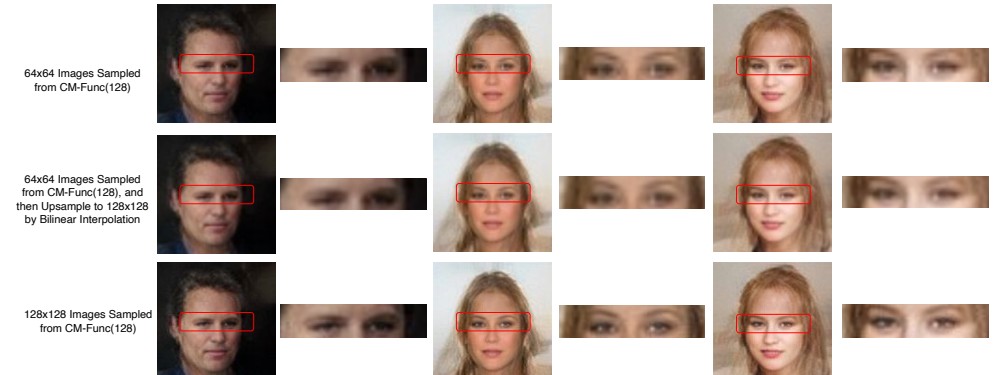

Figure 19: The visual comparison of our image function interpolation results and bi-linear interpolation results from CM-Func trained with 128x128-resolution images. We find that directly sampling 128x128 images from CM-Func has a better visual performance than sampling 64x64 images from CM-Func and then upsampling to 128x128 images by bi-linear interpolation. This result indicates that our CM-Func learns a more complex interpolation mechanism during training compared to the simple linear interpolation mechanism. **(Please see it in color version and zoom in for better visual performance)**

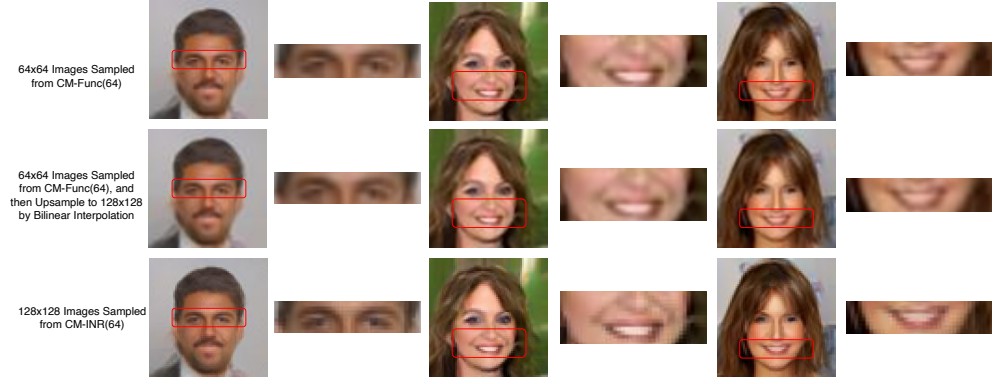

Figure 20: The visual comparison of our image function interpolation results and bi-linear interpolation results from CM-Func trained with 64x64-resolution images. We find that directly sampling 128x128 images from CM-Func (higher than training resolutions) leads to sharper edges than sampling 64x64 images from CM-Func and then upsampling to 128x128 images by bi-linear interpolation. This is because the image function based on INR is a complex continuous function that naturally supports more complex interpolation. **(Please see it in color version and zoom in for better visual performance)**

realistic than those from $64 \times 64$, which indicates that a larger noise space leads to better generation performance. We also present quantitative results on the CelebA-128 dataset with different training resolutions of our function generator in Table 7. We find that even trained with a lower resolution image, our function generator still has a good performance and is better than the U-Net generator. In addition, we present the inference FPS comparison in Figure 17, which shows that denoising from a smaller noise leads to faster inference speed. We also show that our function generator can generate non-squared samples with any height-weight ratio in Figure 18. This process can be easily implemented by querying the image function with coordinates of different height-width ratios.

**Efficient sampling process for multi-resolution images.** Our pipeline is the first work to generate functions instead of images. Based on such generated functions, we can decouple the resolution of generated images and the total amount of the parameters generated from the neural network. Our pipeline only needs to apply the inference forward once to get the image function, which can be rendered into images with any resolution. For example, if we use the traditional U-Net to generate 10000 images, each of which requires 3 different resolutions ($32 \times 32$, $64 \times 64$ and $128 \times 128$), we need to separately apply the inference forward process with 3 different models on 30000 different noise images. On the contrary, our pipeline only needs to generate 10000 functions, each of which will be rendered into three images with negligible render time. As shown in Table 2, our pipeline achieves nearly about 400%+ FPS to sample all these images. Therefore, compared to the U-Net generators, our proposed function generator enjoys much better inference efficiency when generating images with different resolutions.

**Better super-resolution performance than simple linear interpolation.** We show the comparison of the interpolation performance of our image functions or simple linear interpolation in Figure 19 and Figure 20. In the case of sampling resolution no exceeding training resolution, we find that generating samples with 128x128 resolution directly with CM-Func-128 yields much better results than sampling at 64x64 and then upscaling to 128x128 with bilinear interpolation, as CM-Func learns some non-linear interpolation mechanism during optimization. In the case of sampling resolution exceeding training resolution, we find that generating 128x128 samples with CM-Func-64 also produces better visual results than sampling at 64x64 and then upscaling to 128x128 with bilinear interpolation, e.g. sharper edges, because INR inherently provides stronger non-linear interpolation capabilities.

# E  BROADER IMPACTS

Image generation is a widely discussed problem. Deep learning generative models can greatly impact society. Positively, they can enhance creative industries, healthcare, and education by generating art, improving medical imaging, and personalizing learning. However, they also pose risks, such as creating deepfakes, spreading misinformation, and raising ethical concerns about data privacy and bias. Balancing these benefits and risks is essential for their responsible use.