# OpenReview forum: "One-step Image-function Generation via Consistency Training"
_ICLR.cc/2025/Conference — ICLR 2025 Conference Withdrawn Submission_

### Official Review · Reviewer_N1ca · 2024-11-01

**Soundness:** 2
**Presentation:** 2
**Contribution:** 2
**Rating:** 5
**Confidence:** 4

**Summary:**

This paper proposes a novel approach to image generation through consistency models, aiming to improve efficiency in generating high-quality, variable-resolution images. By adopting a Transformer-based generator that leverages implicit neural representations (INRs), the authors propose an architecture allowing flexible resolution generation with reduced resource demands. The method addresses challenges associated with traditional U-Net models by decoupling image resolution from model parameters and incorporating a pre-training phase for enhanced consistency training.

**Strengths:**

1. The introduction of a Transformer-based generator that produces image functions is an efficient approach that enables any-resolution sampling. This is a significant step forward from fixed-resolution U-Net generators.
2. By decoupling image resolution from model parameters, the proposed method reduces computational overhead and GPU memory usage, allowing more accessible high-resolution image generation.
3. The pre-training task effectively enhances the consistency model’s performance, leading to faster convergence and better denoising capabilities compared to models trained from scratch.

**Weaknesses:**

1. The paper utilizes Transformers in a relatively straightforward way for image generation. While the INR-based function generator is effective, the paper could benefit from a clearer explanation of how it fundamentally diverges from other Transformer-based models in diffusion applications.
2. The pre-training phase, while beneficial, adds additional complexity to the training pipeline. It would be helpful to compare the training cost between this method and other approaches.
3. The comparisons with existing one-step diffusion methods are missing. In fact, there are a lot of one-step methods, including ADD and DMD.
4. Given that the method proposed by the authors is capable of generating images of arbitrary resolution, in the selection of datasets in Section 4.1, the authors should consider including more datasets with various resolutions beyond the current 64 and 128 to facilitate a comprehensive comparison. In fact, a larger resolution has become more popular, e.g. 512 and 1024. It is hard to justify whether this method can actually accommodate arbitrary resolution without reporting the results of high resolution image synthesis.
5. To evaluate the method, more metrics should be considered when comparing different methods, including NIQE, CLIPIQA, MUSIQ, LPIPS, MANIQA, DISTS.

**Questions:**

More results are required and more methods should be compared.

---

> ### Author Response · Authors · 2024-11-20
> **Response to Reviewer N1ca (Part I)**
>
> **W1:**  The paper ... applications.
>
> **R1:** Our method fundamentally diverges from other Transformer-based models in diffusion. **The major contribution of this paper is much more than introducing Transformers in consistency models but delivering Transformers to generate an image function**, which is parameterized as the MLPs, as presented in Figure 1. We have provided a detailed discussion of the difference between our model with U-Net and other Transformer-based models in lines 479-504 and Table 2. Specifically, **the traditional consistency pipelines suffer from fixed-resolution image generation**, therefore if they are required to generate images with different resolutions, they cannot generate consistent images across different resolutions. And they are required to train separate models at different resolutions to generate images at different resolutions. The inference process is independent across different resolutions and therefore is very costly (See the Multi-Resolution Sampling Sampling FPS in Table 2). In contrast, our method delivers a Transformer to generate image functions, which can then be rendered into any-resolution images with nearly negligible time cost. Therefore, our method is quite suitable for the scenario where images with multiple resolutions are required.
>
> **W2:** The pre-training ... approaches.
>
> **R2:** Thank you very much for your suggestion.  We need to emphasize that the **training time for pretraining is much less than the training time for consistency training, which is so resource-costly.** Take the CelebA-64 dataset (with a total of 202,599 images) as an example. Pretraining for 30 epochs (with a batch size of 32) requires approximately 190,937 iterations. This is only about one-fifth of the 900,000 iterations required for consistency training. In addition, the pretraining is applied solely to the function prediction module, which accounts for less than half of the total model parameters. As a result, the time spent on pretraining is significantly lower than that of consistency training, amounting to about 5%. The exact training time for different models is presented here:
> ||Total Training Time (GPU * Hours) for CelebA-64|
> |-|-|
> |CM-UNet| 302.5
> |CM-DiT | 207.5
> |CM-UViT| 212.5
> |CM-Func w.o. pretrain| 223.75
> |CM-Func w. pretrain| 223.75 + 12.9
>
> **W3:** The ... DMD.
>
> **R3:** Thank you very much for bringing our attention to these interesting works. We admit that several one-step methods have been proposed. However, **most of these works need to finetune or distill from an existing diffusion model**, e.g. DMD [1] mentioned by you and the CD mode from consistency models. The distillation-based methods may suffer from the bias of the pre-trained diffusion model. In contrast, our method relies on consistency training that can **train the generative model in isolation and only from data**, which should be a more flexible and convenient training mechanism as we mention in lines 40-42 and "as an independent family of generative models" by [2][3]. More importantly, our model **generates an image function that supports any-resolution image generation, and this is not achieved by other one-step diffusion methods.**
>
> **W4:** Given ... synthesis.
>
> **R4:** Thank you very much for your suggestion about training on higher-resolution images. However, it is quite unaffordable to train consistency models in such a high resolution. By now, we have showcased that even though we train our model in a 64 or 128-resolution image dataset, our model can generate consistent any-resolution images, e.g., images with 512 resolution in Figure 9. Since we generate an image function that can be quired with arbitrary coordinates within a continuous range, our model **can theoretically guarantee image generation with arbitrary resolution**. **Training with high resolution can improve the performance of our model on high-resolution image generation (a question about better or worse), but it does not affect the ability that our model can generate arbitrary resolution images (a question about yes or no).**

---

> ### Author Response · Authors · 2024-11-20
> **Response to Reviewer N1ca (Part II)**
>
> **W5:** To ... DISTS.
>
> **R5:** Thank you very much for bringing our attention to these image quality assessment metrics. We mainly follow the evaluation process of current generative models, which is to evaluate the difference between the distribution of training images and generated images. We also follow your suggestion to evaluate different methods in terms of the image quality assessment metrics on two datasets. Since LPIPS and DISTS are full reference metrics, the generative models do not have ground truth, so we cannot evaluate such two metrics. We use the pyiqa package to evaluate the no reference metrics mentioned by you, i.e., NIQE for distorted images, CLIPIQA for visual consistency and comprehensibility of image content, MUSIQ for image structure and distortion in images, and MANIQA for naturalness and structural information of images. Specifically, we follow the setting mentioned in line 401 in the paper to evaluate NIQE, CLIPIQA, and MUSIQ on 50000 generated images and evaluate MANIQA on 1000 generated images for efficiency. The results are provided in the following two tables. **The best results are marked as bold and the second results are marked with underline.**
>
> ||Models|NIQE $\downarrow$|CLIPIQA $\uparrow$|MUSIQ $\uparrow$|MANIQA $\uparrow$
> |-|-|-|-|-|-|
> |Cifar10-32|CM-UNet| 23.003 | 0.516 | 17.043 | 0.105
> |Cifar10-32|**CM-Func**| **22.264** | **0.520** |**17.143** |**0.107**
> |CelebA-64|CM-UNet | 6.810  | 0.465 | 21.198 | 0.210
> |CelebA-64|**CM-Func** | **6.503**  | **0.549** | **22.230** | **0.220**
>
> ||Models|NIQE $\downarrow$|CLIPIQA $\uparrow$|MUSIQ $\uparrow$|MANIQA $\uparrow$
> |-|-|-|-|-|-|
> |CelebA-64|CM-UNet | 6.810 | 0.465 | 21.198 | 0.210
> |CelebA-64|CM-UViT | $\underline{6.564}$  | 0.516 | **22.805** | 0.194
> |CelebA-64|CM-DiT  | 6.666  | **0.561** | 22.139 | **0.234**
> |CelebA-64|CM-Func | **6.503**  | $\underline{0.549 }$| $\underline{22.230}$ | $\underline{0.220}$
>
> **We find that these quantitative results are highly consistent with our conclusions in the paper (lines 499-504), which is 1) our model has a better performance than the original U-Net in terms of all metrics. 2) the performance of our function generator is comparable with the Transformer-based generators.**
>
> [1] Yin, Tianwei, et al. "One-step diffusion with distribution matching distillation." CVPR. 2024.
>
> [2] Song, Yang, et al. "Consistency Models." ICML, 2023.
>
> [3] Song, Yang, and Prafulla Dhariwal. "Improved Techniques for Training Consistency Models." ICLR, 2024.

---

> ### Author Response · Authors · 2024-11-25
>
> Dear Reviewer N1ca,
>
> We appreciate the time and effort that you have dedicated to reviewing our manuscript. We have carefully addressed all your queries. Could you kindly spare a moment (approximately 2 minutes) to review our responses? Have our responses addressed your major concerns? If there is anything unclear, we will address it further. We look forward to your feedback.
>
> Best regards,
>
> Authors of Paper 9675

---

> ### Author Response · Authors · 2024-11-26
>
> Dear Reviewer N1ca,
>
> We appreciate the time and effort that you have dedicated to reviewing our manuscript. We have carefully addressed all your queries. Could you kindly spare a moment (approximately 2 minutes) to review our responses? Have our responses addressed your major concerns? If there is anything unclear, we will address it further. We look forward to your feedback.
>
> Best regards,
>
> Authors of Paper 9675

---

> > ### Comment · Reviewer_N1ca · 2024-11-28
> > **Thanks for the response**
> >
> > After reading the response, my major concerns are still not well addressed.
> >
> > First, the comparisons with one-step diffusion models are still missing. I agree that there are some differences among these methods. Nevertheless, it does not mean the results of these methods cannot be compared. Moreover, the authors claimed that "distillation-based methods suffer from the bias of the pre-trained diffusion model". It is unclear what the bias is exactly. It would be better to clarify it.
> >
> > Second, "any resolution" seems to be overclaimed. The results in the resolutions higher than 128x128 are still missing. Although any resolution generation is possible, it is hard to believe whether this method really works well in high resolution.
> >
> > Best,
> >
> > Reviewer N1ca

---

> > > ### Author Response · Authors · 2024-11-28
> > >
> > > >First, the comparisons with one-step diffusion models are still missing. I agree that there are some differences among these methods. Nevertheless, it does not mean the results of these methods cannot be compared. Moreover, the authors claimed that "distillation-based methods suffer from the bias of the pre-trained diffusion model". It is unclear what the bias is exactly. It would be better to clarify it.
> > >
> > > **R6:** Thank you very much for your reply. We claim that "distillation-based methods suffer from the bias of the pre-trained diffusion model" because **these models require a pre-trained diffusion model to estimate the PF ODE trajectories.** For example, CD needs a one-step ODE solver based on a pre-trained diffusion model to generate a pair of adjacent data points on the PF ODE trajectory. The distillation indeed eases the difficulty of estimating the PF ODE trajectories in the early stage of training, which leads to a more efficient training process and slightly better performance than CT as presented in the original CM paper [1].
> > > However, the estimated PF ODE trajectories fairly contain some errors (or bias) from the real PF ODE trajectories, which leads to a problem recognized by researchers that **distillation limits the sample quality of the resulted model to that of the distillated diffusion model**, i.e., "distillation limits the sample quality of the consistency model to that of the diffusion model" from [2]. In contrast, our model relies on the consistency training target that learns the PF ODE trajectories from pure training data. Specifically, the consistency training uses the Euler method as the ODE solver as an unbiased estimation in the limit of the number of discretized interval $N \rightarrow \infty$ [1]. The consistency training should be a more flexible and convenient training mechanism and "as an independent family of generative models" by [1][2]. Therefore, we focus our comparisons on **the different network architectures within the consistency training framework**. Besides the baseline U-Net model implemented in the original paper, we also implemented the consistency training with **two popular architectures (DiT and UViT) and presented lots of discussions with them.** We believe we have presented sufficient experiments for comparisons to support our major claims.
> > >
> > > > Second, "any resolution" seems to be overclaimed. The results in the resolutions higher than 128x128 are still missing. Although any resolution generation is possible, it is hard to believe whether this method really works well in high resolution.
> > >
> > > **R7:** Thank you for your reply. We have provided **sufficient evidence in the paper that our model is able to generate any-resolution images, including high-resolution images, even though it is trained in a relatively low-resolution dataset,** e.g., Figure 9, 18, 19, 20. The inference resolution in Figure 9 is up to 512 (of course it can be higher).
> > >
> > > We need to clarify **the difference between the contribution of "any-resolution" and "high-resolution" since the target of this paper is to propose a method for any-resolution image generation.** Any-resolution image generation has extensive usage scenarios, e.g., device adaptation and network bandwidth & load time optimization.
> > >
> > > In device adaptation,  by leveraging a single model capable of generating images at varying resolutions, it seamlessly adapts to devices with different display capabilities, ranging from mobile devices to high-definition monitors. **Unlike traditional methods that require multiple models or specific versions tailored to each device, our approach eliminates the need for redundant training processes, significantly reducing storage and computational overhead.**
> > >
> > > From a network bandwidth perspective, **the model dynamically adjusts the output resolution based on available network conditions, making it particularly well-suited for real-time applications in environments with varying bandwidth.** By generating lower-resolution images for devices or connections with limited bandwidth, the model ensures faster load times and smoother user experiences, without compromising the quality of higher-resolution images when the network and device capabilities allow. This feature is especially beneficial in mobile and cloud-based applications where data transfer constraints and latency are critical considerations.
> > >
> > > As a result, **the "any-resolution" feature of our approach not only optimizes resource utilization but also enhances scalability and adaptability across different devices and network conditions, providing a more efficient and flexible potential solution for dynamic image generation tasks. This contribution is totally different from those methods that target the "high-resolution" image generation with very good quality.**
> > >
> > >
> > > [1] Song, Yang, et al. "Consistency Models." ICML, 2023.
> > >
> > > [2] Song, Yang, and Prafulla Dhariwal. "Improved Techniques for Training Consistency Models." ICLR, 2024.

---

> ### Author Response · Authors · 2024-12-01
>
> Dear Reviewer N1ca,
>
> We appreciate the time and effort that you have dedicated to reviewing our manuscript. We have carefully addressed all your queries and uploaded a new PDF version. Have our responses addressed your major concerns? If you have further concerns, please discuss them with us. We will address it further. We look forward to your feedback.
>
> Best regards,
>
> Authors of Paper 9675

---

### Official Review · Reviewer_NMKa · 2024-11-02

**Soundness:** 3
**Presentation:** 2
**Contribution:** 2
**Rating:** 5
**Confidence:** 3

**Summary:**

The paper addresses the limitations of using a U-Net generator with consistency models, i.e., the substantial computational resources required and the difficulty in generating images at user-specified resolutions. To address these challenges, the researchers propose replacing the U-Net generator with an implicit neural representation (INR), which demonstrates potential in producing images with scalable resolutions. The proposed method reduces training costs relative to the U-Net generator while achieving superior image quality as quantified by common evaluation metrics.

**Strengths:**

The proposed method is conceptually sound and effectively addresses the limitations of the U-Net-based consistency model.

Experimental results support the efficacy of incorporating INR within consistency models for improved image generation.

**Weaknesses:**

(1) Novelty: while INR is applied here for high-resolution image generation within the context of consistency models, INR is already a widely-used technique in other 2D image generation frameworks. The contribution appears to be somewhat incremental.

(2) Related work: the review of related work on INR-based methods is somewhat insufficient, particularly in the context of high-resolution image generation. Additional discussion on alternative high-resolution generation strategies would be beneficial.

(3) Performance comparison: although the method shows reduced computational cost, its image quality appears less competitive compared to replacing UNet with DiT, as observed in Table 2.

(4) Computation cost comparison: it would be helpful to include a broader computational cost comparison with other methods listed in Table 2, rather than restricting comparisons solely to the CM-UNet model.

(5) It would be clearer and more concise to use “Eq.” rather than “Eq. equation.” when referring to equations.

**Questions:**

See the limitations above, which detail the questions concerned, and it is expected to address these issues.

---

> ### Author Response · Authors · 2024-11-20
> **Response to Reviewer NMKa (Part I)**
>
> **W1:** (1) Novelty ... incremental.
>
> **R1:** We need to emphasize that we deliver INR in consistency models mainly for **one-step any-resolution image generation**, which mainly targets the efficient generation problem when generating images with multiple resolutions or unknown arbitrary resolutions.
>
> Our paper shows several differences and advantages with other image generation frameworks with INRs, and the greatest advantages are the **flexible and efficient one-stage end-to-end training pipeline and the one-step generation process**.  As we have discussed in "diffusion models based on implicit neural representations" in the related work part of the paper (lines 162-169), most works that apply INR in their works are **two-stage training pipelines**. They consider the process of converting signals to the INR (encoding signals to the INR space) and the process of diffusion on the INR space as two processes that are totally independent of each other. The two-stage approach helps to ease the difficulty of generating INRs but the training process is inflexible and the error in the first representation stage would greatly affect the performance of the second diffusion stage, e.g. only 40.40 FID reported by Functa in CelebA-HQ 64^2 dataset. In contrast, our method enjoys the **one-stage end-to-end training process** and optimizes all modules from pure data.
>
> Naively implementing a one-stage training pipeline for diffusion on INRs **requires evaluating the denoiser network in the image space and will face two challenges.** 1) it is unaffordable to evaluate the denoiser network in the image space for diffusion models because they need to render the generated INRs into images for each diffusion step, which makes the inference process very costly. 2) It is very hard for a denoiser network to generate INRs for images with so much noise because INRs prior to fit low-frequency signals and are hard to fit high-frequency noises. Therefore, they deliver a two-stage training pipeline to avoid these two challenges: firstly converting all signals to their INRs and directly applying diffusion on the INR space.
>
> In contrast, **our method relying on consistency training has many insights and solves these two challenges in a much clever way**. 1) The consistency training enables our model to generate INRs with just a single diffusion step, therefore it is affordable for us to directly evaluate the denoiser network in the image space. 2) The target of consistency training is to train a denoiser network to map all points in the PF-ODE trajectory into the original image, therefore our denoiser network only needs to generate INRs for images with little noise, which greatly improves the training efficiency of the denoiser network.
>
> As a result, we believe **our method is novel and should be distinguished from other INR-based diffusion models.**
>
> **W2:**  (2) Related work... beneficial.
>
> **R2:** Thank you very much for the suggestion about additional discussion with high-resolution generation strategies. Most high-resolution generation methods are based on generation in a patch-by-patch manner, e.g. [1] suggested by Reviewer 8PPP. However, our method focuses on generating the entire image function in a single step, and the image function is represented by a global function and can be rendered into any-resolution images. Basically, **the patch-by-patch approach for generating high-resolution images is orthogonal to our method of generating the entire image function as a whole**. The former focuses more on generating local regions iteratively, while our method emphasizes the global representation. Additionally, it is entirely feasible to combine our method with the patch-by-patch strategy. Specifically, we can generate a global signal within each patch to enable fast generation with flexible resolutions at the patch level, and then use the patch-by-patch approach to assemble a high-resolution image.
>
> **W3:** (3) Performance ... Table 2.
>
> **R3:** We need to clarify that **our model is comparable with Transformer-based generators, e.g. better IS metric than DiT.** Table 2 is mainly for the ablation study. It shows that DiT should be a better denoiser for consistency models than the original U-Net, which motivates us to deliver a DiT-like encoder as our feature extraction module that handles input noisy images and modulates noise level embedding. However, as we have discussed in Lines 479-504, our method generates image function rather than directly generating fixed-resolution images. **The architectures (U-Net, UViT, DiT) that generate fixed-resolution images** suffer from multiple training and inference processes when generating images with multiple resolutions, therefore they are less flexible and have a much lower multi-resolution sampling FPS than our method.

---

> ### Author Response · Authors · 2024-11-20
> **Response to Reviewer NMKa (Part II)**
>
> **W4:** (4) Computation...model.
>
> **R4:** Thank you very much for your suggestion. We have shown in lines 847-849 that we implemented the DiT and UViT with a close amount of learnable parameters with our function generator. So **the parameter amount and GPU cost of DiT and UViT are very close to our function generators**. We will highlight this setting in our future version. The major difference in the computation cost comparison between DiT, UVit, and our function generator is the multi-resolution sampling FPS, which has been shown in Table 2.
>
> **W5:** (5) It ... equations.
>
> **R5:** Thank you very much for pointing out our typo.
>
>
> [1] Any-resolution training for high-resolution image synthesis, ECCV 2022

---

> > ### Comment · Reviewer_NMKa · 2024-11-23
> > **Thank authors for the response**
> >
> > Thank the authors for the response. I believe that incorporating INR into the consistency model can offer potential benefits, such as enabling generation at arbitrary resolutions. Thank the authors for clarifying the structural differences between this work and other INR-based approaches. However, I would suggest focusing the contribution on a major point — whether that is any-resolution generation, high-quality generation, or computational efficiency.
> >
> > Currently, the comparisons utilize different baselines for various aspects. While the results show certain benefits, it is challenging to see a clear advantage in any particular area. For instance, regarding any resolution generation, a systematic review and evaluation are not clearly presented. When considering generation quality, the improvement reported in Table 2 and in response to reviewer N1ca does not appear clear compared to the CM-DiT baseline. Other reviewers seem to have similar concerns. I recommend emphasizing one primary contribution with clear, compelling advantages, supported by systematic comparisons and evaluations.

---

> > > ### Author Response · Authors · 2024-11-23
> > > **Response to Reviewer NMKa**
> > >
> > > > However, I would suggest focusing the contribution on a major point — whether that is any-resolution generation, high-quality generation, or computational efficiency.
> > >
> > > **R6:** Thank you very much for your reply. In fact, in the very beginning of the paper, i.e. (b)(c) in Figure 1 and lines 94-95, we have clearly shown that our method enjoys a great contribution that can distinguish our method from other methods, which is **the efficient any-resolution image generation**, especially when generating images with multiple resolutions or unknown resolution.
> > >
> > > > Currently, the ... are not clearly presented.
> > >
> > > **R7:** Thank you for your suggestion. We did not compare our method to existing any-resolution generation methods because **the efficiency of these methods is theoretically not on the same order of magnitude as ours, especially the efficiency when scaling to higher resolution images. We then focus on discussing the efficiency of our method in comparison to consistency models, which are highly efficient in the existing generative models.**
> > >
> > > **Current papers on any-resolution image generation, such as [1][2], do not provide any discussions about their generation efficiency.** This is because these methods inherently require a long time to generate any resolution images, either through patch-by-patch generation [1] or two-stage INR-based diffusion [2]. The former requires multiple forward passes to generate many patches, while the latter necessitates a large number of diffusion steps.
> > >
> > > We test the inference efficiency for the patch-by-patch method [1] and find that it takes about 0.04s to generate a 256\*256 patch, therefore [1] has about 25 FPS for 256\*256 images and 6.25 FPS for 512\*512 images. **The inference efficiency is linearly decreasing as the resolution increases.** However, the inference efficiency of our model is much less affected by the resolution of generated images, because our model only needs to generate fixed shapes for the parameters of the image function and the process of querying pixels for each coordinate can be parallel.
> > >
> > > ||Inference FPS-256 $\uparrow$ |Inference FPS-512$\uparrow$ | FPS Scaling down Rate from 256 to 512 $\downarrow$
> > > |-|-|-|-|
> > > |anyres-GAN [1]  | 25 | 6.25 | 25/6.25=4
> > > |Ours | **96** | **71** | **1.35**
> > >
> > > Therefore, **in the paper, we focus on discussing the efficiency of our method in comparison to consistency models, which are highly efficient in the existing generative models.** Consistency models based on U-Net can achieve highly efficient one-step diffusion, and our method further enhances this by 1) incorporating a more efficient network architecture that improves generation quality and computational efficiency, and 2) providing the ability to generate any-resolution images, making consistency models even more efficient.
> > >
> > > > When considering ... evaluations.
> > >
> > > **R8:** Thank you for your question. We claim that **our model is comparable with Transformer-based generators. We believe this claim should be well supported by Table 2 and the quantitative result of the IQA metrics reported in the response to reviewer N1ca.** In most metrics, our method achieves comparable or even better results than CM-DIT, such as IS and NIQE.
> > >
> > > Basically, it is reasonable to observe some quality performance drop if we force the network to generate an image function rather than an image. This is because directly generating an image does not require modality transformation. The network's input and output are exactly the same shape, so it is easy to use residual connections to connect the input and output and denoise the input signal.  However, outputting an image function involves modality transformation (image to INR), and the network needs to achieve both denoising and modality transformation functions, which makes it harder to train a unified network to achieve both functions.
> > >
> > > In fact, we present the quantitative results in Table 2 to show the effectiveness of our  DiT-like encoder in our feature extraction module. **Even though generating an image function brings some difficulties to our network training, our function generator can still achieve comparable performance with those image generators.**
> > >
> > > In addition, our function generator **enjoys a more flexible sampling process and higher multi-resolution sampling FPS than the traditional image generators.**
> > >
> > > [1] Any-resolution training for high-resolution image synthesis, ECCV 2022
> > >
> > > [2] Image Neural Field Diffusion Models, CVPR 2024

---

> ### Author Response · Authors · 2024-12-01
>
> Dear Reviewer NMKa,
>
> We appreciate the time and effort that you have dedicated to reviewing our manuscript. We have carefully addressed all your queries and uploaded a new PDF version. Have our responses addressed your major concerns? If you have further concerns, please discuss them with us. We will address it further. We look forward to your feedback.
>
> Best regards,
>
> Authors of Paper 9675

---

> > ### Comment · Reviewer_NMKa · 2024-12-01
> > **Responce to author**
> >
> > Thanks for the author's rebuttal. If, as responded by the author, the core innovation of this paper is "any-resolution image generation," I believe it should be compared with existing diverse methods for any-resolution generation in terms of generation performance, rather than efficiency. However, such a comparison is currently missing. Moreover, the generation performance of the proposed method is only comparable to that of simply replacing UNet with a Transformer-based generator, without demonstrating the performance advantages of INR-based consistency training. While I appreciate the author's detailed responses, I believe the core issue has not been addressed. Therefore, my final decision is to reject the paper.

---

### Official Review · Reviewer_ftNQ · 2024-11-03

**Soundness:** 2
**Presentation:** 2
**Contribution:** 2
**Rating:** 5
**Confidence:** 4

**Summary:**

This paper addresses two main issues: the training instability of consistency models with small batch sizes and the limitation of generating images at fixed resolutions. To tackle these challenges, the authors propose using a Transformer-based generator along with implicit neural representations. Additionally, to improve training stability, they introduce an auxiliary task before training the consistency model, which leads to faster convergence and enhanced image generation quality. Experimental results show that this approach improves performance in one-step image generation with reduced training requirements and enables efficient, any-resolution image sampling.

**Strengths:**

A key strength of this paper lies in its innovative design of a consistency model that supports multi-resolution sampling, overcoming the fixed-resolution limitations of traditional models. The approach also effectively addresses training instability at low batch sizes, making it feasible to train with fewer resources.

**Weaknesses:**

While the paper presents improved training efficiency as a key contribution, there are two aspects that raise questions regarding this claim:

1. In comparison with Song et al.’s experimental setup, it seems expected that training with a smaller batch size would lead to lower performance. To convincingly demonstrate an improvement in training efficiency, comparing the proposed model with a consistency model trained on low batch sizes may be insufficient. Instead, it would strengthen the argument to show that the proposed method performs better than models trained with larger batch sizes.
2. In Figure 8, it appears that pre-training is essential for reaching the convergence point of “Denoising Distance.” However, considering the overall training time, if an additional 30 epochs of pre-training are required compared to traditional methods, it may be worth questioning whether this approach can truly be considered efficient.

**Questions:**

1. Total Training Time: Could the authors clarify the total training time required for the model? While it is mentioned that 30 epochs were used for pre-training, it would be helpful to know the training duration for both CM-UNet and CM-Func models.

2. Evaluation in Table 2: The evaluation process for Table 2 is unclear, particularly regarding how the multi-resolution sampling FPS was measured for CelebA 64. Additional explanation on the methodology used for this metric would be appreciated.

3. Effectiveness with Larger Batch Sizes: It would be interesting to know if the proposed method continues to perform better than models trained with batch sizes larger than 32.

4. Related Works : Adding the following reference to the related work section would enhance the context of the study: Zhuang, Peiye, et al. "Diffusion probabilistic fields." The Eleventh International Conference on Learning Representations, 2023.

---

> ### Author Response · Authors · 2024-11-20
> **Response to Reviewer ftNQ**
>
> **W1**: In comparison ... sizes.
>
> **R1**:  We have shown that the relatively high FID values are due to the low training batch size (32) that allows training with a single GPU. In fact, we have tried training with more GPUs and larger batch sizes (64 and 128). **The claim that the consistency training with UNet will oscillate is consistent as the training batch size reaches 128**, which harms the performance of consistency training. In contrast, our CM-func can have a more stable consistency training process which leads to better generation quality. It is unaffordable for us to train the model exactly following Song et al.’s experimental setup which has a batch size of 2048 or 4096.
>
> **W2**: In Figure 8, ..., efficient.
>
> **R2**: We need to emphasize that the **training time for our pretraining step is much less than the training time for consistency training, which is so resource-costly.** Take the CelebA-64 dataset (with a total of 202,599 images) as an example. Pretraining for 30 epochs (with a batch size of 32) requires approximately 190,937 iterations. This is only about one-fifth of the 900,000 iterations required for consistency training. In addition, the pretraining is applied solely to the Function Prediction Module, which accounts for less than half of the total model parameters. As a result, the time spent on pretraining is significantly lower than that of consistency training, amounting to about 5%. The exact training time for different models is presented here:
>
> ||Total Training Time (GPU * Hours) for CelebA-64|
> |-|-|
> |CM-UNet| 302.5
> |CM-DiT | 207.5
> |CM-UViT| 212.5
> |CM-Func w.o. pretrain| 223.75
> |CM-Func w. pretrain| 223.75 + 12.9
>
> **Q1:**  Total Training Time.
>
> **R3**:  Please check the above table.
>
> **Q2:** Evaluation in Table 2.
>
> **R4**: We apologize for the unclear caption for Table 2. It should be "Table for efficiency and accuracy of different generators on CelebA dataset". The Accuracy is evaluated on the CebebA dataset with resolution of 64.
>
> And the multi-resolution sampling FPS is described in lines 483-485 and line 495. We use this metric to **evaluate the efficiency of different models if they are used to generate images with different resolutions**. In our setting, it is calculated as the FPS that generates 10000 image functions, each of which requires 3 different resolutions ($32 \times 32$, $64 \times 64$, $128 \times 128$, therefore a total of 30000 images). The traditional models (CM-UNet, CM-UViT, CM-DiT) cannot sample any-resolution images, so they are required to train three separate models for three different resolutions. The multi-resolution sampling FPSs for these three models are calculated as $\frac{30000}{\sum_{i=1}^{10000}{T_{32}^i}+\sum_{i=1}^{10000}{T_{64}^i}+\sum_{i=1}^{10000}{T_{128}^i}}$, where $T_k^i$ is the average time that generates $i^{th}$ image with resolution $k$. For our CM-Func, we only need to generate one image function for each image and then can be rendered as any-resolution images, so the multi-resolution sampling FPS for CM-Func can be calculated as $\frac{30000}{\sum_{i=1}^{10000}{(T^i}+T_{R32}+T_{R64}+T_{R128})}$, where $T^i$ is the average time that generates $i^{th}$ image functions, and $T_{R32}, T_{R64}, T_{R128}$ are the time to render the image function to image with 32/64/128 resolutions (which is nearly negligible compared to $T^i$). This metric reflects the efficiency of sampling any-resolution images of our method compared to traditional models.
>
> **Q3:** Effectiveness with Larger Batch Sizes.
>
> **R5:** Yes. In fact, we have tried training with more GPUs and larger batch sizes (64 and 128). **The claim that the consistency training with UNet will oscillate is consistent as the training batch size reaches 128**, which harms the performance of consistency training.
> Our model continues to perform better than the original CM at these batch-size settings. Please check the R1 response.
>
> **Q4:** Related Works.
>
> **R6:** Thank you very much for bringing our attention to this interesting work. We will add this reference to our related works. Different from our work, this work delivers explicit field characterization to model signals with different modalities (images, shapes, and spherical data). Their target is to provide a unified framework that can denoise on the field with a single training stage compared to Functa and GEM. Due to the explicit field characterization, **their model cannot achieve any-resolution image sampling as their coordinate is fixed during training**. In contrast, the models relying on implicit fields have the potential for any-resolution image sampling, such as Functa and our method. As discussed in related work in our paper, Functa requires two-stage training while our model can be trained within a end-to-end manner. In addition, our model can achieve one-step generation due to the consistency training while DPF and Functa still require multiple-step sampling because their models are implemented in DDPM.

---

> ### Author Response · Authors · 2024-11-25
>
> Dear Reviewer ftNQ,
>
> We appreciate the time and effort that you have dedicated to reviewing our manuscript. We have carefully addressed all your queries. Could you kindly spare a moment (approximately 2 minutes) to review our responses? Have our responses addressed your major concerns? If there is anything unclear, we will address it further. We look forward to your feedback.
>
> Best regards,
>
> Authors of Paper 9675

---

> > ### Comment · Reviewer_ftNQ · 2024-11-25
> >
> > Thank you for providing clarification on the questions I raised. I appreciate the effort in addressing my concerns. The proposal of a consistency model capable of generating at arbitrary resolutions indeed shows potential benefits. However, I still have concerns regarding the performance of the proposed methodology. While I acknowledge the stability your approach offers for training in low-resource scenarios, I believe it is important to compare the quality of generated images directly with a consistency model trained using existing methods.
> >
> > For example, referring to the results in Song et al. (2023), a consistency model trained on CIFAR-10 achieved an FID of 8.70 for 1-step generation. Demonstrating that your approach achieves performance at least comparable to, if not better than, these results would significantly strengthen your claims.
> >
> > For these reasons, I have decided to maintain my score. Thank you again for your responses and the additional insights.
> >
> > Reference:
> > Song, Yang, et al. "Consistency Models." arXiv preprint arXiv:2303.01469 (2023).

---

> > > ### Author Response · Authors · 2024-11-25
> > >
> > > >  I believe it is important to compare the quality of generated images directly with a consistency model trained using existing methods. For example, referring to the results in Song et al. (2023), a consistency model trained on CIFAR-10 achieved an FID of 8.70 for 1-step generation. Demonstrating that your approach achieves performance at least comparable to, if not better than, these results would significantly strengthen your claims.
> > >
> > > **R7:** Thank you very much for your reply. We totally agree with your opinion that it is important to compare the quality of generated images directly with consistency models. And in fact this is exactly how we evaluate our method in our paper.
> > >
> > > As presented in lines 376-377 and line 388, we **exactly follow the setting from Song et al. (2023), except setting the batch size to 32 which is much smaller than Song et al. (2023)**, i.e., 512 for CIFAR-10. We try to train the consistency models with a single or just a few GPUs we can afford.  The results show that the original consistency models with U-Net oscillate when training with small batch sizes, which leads to poor performance, as shown in Figure 1 (a) and Table 1. And we empirically verify that **a Transformer-based generator (including DiT and our function generator) is much more stable, achieving better performance than the U-Net generator.**
> > >
> > > If you have further questions, please feel free to discuss with us.

---

> ### Author Response · Authors · 2024-12-01
>
> Dear Reviewer ftNQ,
>
> We appreciate the time and effort that you have dedicated to reviewing our manuscript. We have carefully addressed all your queries and uploaded a new PDF version. Have our responses addressed your major concerns? If you have further concerns, please discuss them with us. We will address it further. We look forward to your feedback.
>
> Best regards,
>
> Authors of Paper 9675

---

### Official Review · Reviewer_8PPP · 2024-11-04

**Soundness:** 3
**Presentation:** 3
**Contribution:** 3
**Rating:** 6
**Confidence:** 3

**Summary:**

The authors observe that with low training resources and small batch size, the training of UNet-based consistency model is unstable, and proposed a Transformer-based generator that generates network parameters as INR for consistency training. The authors show better training stability and lower FID metric than the original UNet-based consistency model in the low-resource training setting.

**Strengths:**

**Update after rebuttal:**

```
The authors have addressed my questions and I will keep my rating. The idea of using any-resolution representation for consistency models is interesting, while I agree with other reviewers that more solid comparisons to exsiting methods could be helpful (for efficient training or for any-resolution generation).

```

---

1. Using consistency training for image function generation is an interesting direction to explore. Since INRs are any-resolution decoders, it is natural to compute the consistency objective in the rendered patches.
2. The proposed reconstruction pre-training is simple and effective.
3. The authors show improved FID and other metrics. The training stability is also improved compared to the baseline UNet on common datasets.

**Weaknesses:**

1. It is not very clear that why modeling as INR can help improve the stability in low-resource training. With the Transformer generator and INR representation, is the input noisy image / target in training at fixed resolution or varied resolutions? More discussions about the intuition for the improvement might be helpful. Can the reconstruction pre-training also be applied for the UNet consistency model?
2. Despite showing many metrics, the FID values for both the baseline and proposed method are very high (though it is due to the training budget). The results will be more convincing and solid when the methods can achieve a generally better quality.
3. The claim the advantage of any-resolution generation, it is better to discuss and compare to more recent works that specifically works on any-resolution image generation, for example [1, 2].

[1] Any-resolution training for high-resolution image synthesis, ECCV 2022

[2] Image Neural Field Diffusion Models, CVPR 2024

**Questions:**

It is shown in the supplementary that the generated INR has better quality than bilinear interpolation when decoding to high resolutions. Is the high resolution higher than the resolution in training? If it is the case of resolution extrapolation, is any artifact observed in high resolutoins?

---

> ### Author Response · Authors · 2024-11-20
> **Response to Reivewer 8PPP**
>
> **W1**: It is not very clear that why modeling as INR can help improve the stability in low-resource training. With the Transformer generator and INR representation, is the input noisy image/target in training at fixed resolution or varied resolutions? More discussions about the intuition for the improvement might be helpful. Can the reconstruction pre-training also be applied for the UNet consistency model?
>
> **R1**: We attribute the stable training to the **design of the Feature Extraction Module which is based on DiT** that delivers an adaLN-Zero layer to modulate the Transformer encoder. The original DiT for diffusion models has already been shown to exhibit superior scalability compared to U-Net. We adapt it to consistency models and the ablation study for CM-DiT (Table 2) shows that DiT has a consistent performance on consistency models (better than CM-UNet).
> The input noisy image and the target are of a fixed resolution, and the model outputs an image function, which can then be sampled at arbitrary resolutions. In Appendix C.3.2 and Table 4, we have discussed scenarios where the input noise resolution varies for the CelebA dataset. We find that **increasing the resolution of the input noise slightly improves the FID of the generated images due to larger noise space, though this comes at the cost of longer runtime.**
> The **reconstruction pre-training cannot be applied to the UNet consistency model** as the reconstruction pre-training is to train a model that can predict its INR for an input image, while the UNet cannot perform such a function.
>
> **W2**: Despite showing many metrics, the FID values for both the baseline and proposed method are very high (though it is due to the training budget). The results will be more convincing and solid when the methods can achieve a generally better quality.
>
> **R2**: Thanks for your suggestion. We have shown that the relatively high FID values are due to the low training batch size (32) that allows training with a single GPU. In fact, we have tried training with more GPUs and larger batch sizes (64 and 128). **The claim that the consistency training with UNet will oscillate is consistent as the training batch size reaches 128**, which harms the performance of consistency training. In contrast, our CM-func can have a more stable consistency training process which leads to better generation quality.
>
> **W3**: The claim the advantage of any-resolution generation, it is better to discuss and compare to more recent works that specifically works on any-resolution image generation, for example [1, 2].
>
> **R3**: Thank you very much for bringing us to these two interesting works and we will include them in our related work. [1] tries to generate any-resolution images with the manner of patch-by-patch with the supervision of GAN, which is totally different from our setting.  **The output of their generator is still of a fixed resolution**, specifically
> $𝑝 \times 𝑝$, and must be square-shaped. In other words, to generate larger images, their model requires multiple runs to generate patches, which are then concatenated together. In contrast, our model only needs a single run to produce a continuous function of the corresponding image, which can then be sampled at any resolution. Furthermore, **their model cannot generate images at lower resolutions**, as the patches they generate are at least $𝑝 \times 𝑝$. Our model, however, allows for sampling at any desired resolution, offering greater sampling flexibility.
>
> [2] is an interesting work that should be included in the "Diffusion Models Based on Implicit Neural Representations" section of our related work. Same as other diffusion models based on INRs, they first train an INR converter as an encoder module for represetations, and then separately train a diffusion model on these INR representations. **It is a two-stage training and inference pipeline, causing inflexible training and potential error accumulation for each stage.** In contrast, our paper proposes a novel unified architecture that is trained in an end-to-end manner and can generate image functions from noise in a single stage.
>
> **Q1**: It is shown in the supplementary that the generated INR has better quality than bilinear interpolation when decoding to high resolutions. Is the high resolution higher than the resolution in training? If it is the case of resolution extrapolation, is any artifact observed in high resolutoins?
>
> **R4**: Yes, the high resolution sampled from our generator can be higher than the resolution in training. **No artifact is observed.** Here, we follow [3] to apply variational coordinates to eliminate artifacts.
>
>
> [1] Any-resolution training for high-resolution image synthesis, ECCV 2022
>
> [2] Image Neural Field Diffusion Models, CVPR 2024
>
> [3] Attention Beats Linear for Fast Implicit Neural Representation Generation, ECCV 2024

---

> ### Author Response · Authors · 2024-11-25
>
> Dear Reviewer 8PPP,
>
> We appreciate the time and effort that you have dedicated to reviewing our manuscript. We have carefully addressed all your queries. Could you kindly spare a moment (approximately 2 minutes) to review our responses? Have our responses addressed your major concerns? If there is anything unclear, we will address it further. We look forward to your feedback.
>
> Best regards,
>
> Authors of Paper 9675

---

> ### Author Response · Authors · 2024-11-26
>
> Dear Reviewer 8PPP,
>
> We appreciate the time and effort that you have dedicated to reviewing our manuscript. We have carefully addressed all your queries. Could you kindly spare a moment (approximately 2 minutes) to review our responses? Have our responses addressed your major concerns? If there is anything unclear, we will address it further. We look forward to your feedback.
>
> Best regards,
>
> Authors of Paper 9675

---

> ### Author Response · Authors · 2024-12-01
>
> Dear Reviewer 8PPP,
>
> We appreciate the time and effort that you have dedicated to reviewing our manuscript. We have carefully addressed all your queries and uploaded a new PDF version. Have our responses addressed your major concerns? If you have further concerns, please discuss them with us. We will address it further. We look forward to your feedback.
>
> Best regards,
>
> Authors of Paper 9675

---

### Author Response · Authors · 2024-11-27
**For all reviewers**

Dear all reviewers:

Thank you very much for your detailed review. Your constructive suggestions polish our paper a lot. Based on the discussion, we have added more details to our paper and uploaded a new version. The modified part is marked as orange. Please check the PDF and the Supplementary material for the full version.

If you have any concerns, please discuss them with us. We will address them further. We look forward to your feedback.

Best regards,

Authors of Paper 9675

---

### Note · Authors · 2024-12-13

I have read and agree with the venue's withdrawal policy on behalf of myself and my co-authors.